



# Bottom-water deoxygenation at the Peruvian Margin during the last deglaciation recorded by benthic foraminifera

Zeynep Erdem[1], Joachim Schönfeld[2], Anthony E. Rathburn[3], Maria-Elena Pérez[4], Jorge Cardich[5], Nicolaas Glock[2]

[1] NIOZ Royal Netherlands Institute for Sea Research, and Utrecht University, P.O. Box 59, 1790 AB Den Burg, Texel, the Netherlands
[2] GEOMAR Helmholtz Centre for Ocean Research Kiel, Wischhofstr. 1-3, 24148, Kiel, Germany.
[3] Department of Geological Sciences, California State University, Bakersfield, CA, 93311, USA.
[4] Natural History Museum, Department of Palaeontology, London, UK.
[5] Instituto del Mar del Peru (IMARPE), A. Gamarra y Gral. Valle, Chucuito, Callao 01, Peru.

*Correspondence to*: Zeynep Erdem (zeynep.erdem@nioz.nl)

**Abstract.** Deciphering the dynamics of dissolved oxygen in the mid-depth ocean during the last deglaciation is essential to understand the influence of climate change on modern oxygen minimum zones (OMZs). Many paleo-proxy records from the Eastern Pacific Ocean indicate an extension of oxygen depleted conditions during the deglaciation but the degree of deoxygenation has not been quantified to date. The Peruvian OMZ, one of the largest OMZs in the world, is a key area to monitor such changes in near-bottom water oxygenation in relation to changing climatic conditions. Here, we analysed the potential to use the composition of foraminiferal assemblages from the Peruvian OMZ as a quantitative redox-proxy. A multiple regression analysis was applied to a joint dataset of living (rose Bengal stained, fossilizable calcareous species) benthic foraminiferal distributions from the Peruvian continental margin. Bottom-water oxygen concentrations ($[O_2]_{BW}$) during sampling were used as dependant variable. The correlation was significant ($R^2 = 0.82$; $p<0.05$) indicating that the foraminiferal assemblages are rather governed by oxygen availability than by the deposition of particulate organic matter ($R^2=0.53$; $p=0.31$). We applied the regression formula to four sediment cores from the northern part of the Peruvian OMZ between 3°S and 8°S and 600 m to 1250 m water depths; thereby recording oxygenation changes at the lower boundary of the Peruvian OMZ. Each core displayed a similar trend of decreasing oxygen levels since the Last Glacial Maximum (LGM). The overall $[O_2]_{BW}$ change from the Last Glacial Maximum and the Holocene was constrained to 30 μmol/kg at the lower boundary of the OMZ, whereas at shallower depths $[O_2]_{BW}$ was relatively stable along the deglaciation. The deoxygenation trend was time-transgressive. It commenced at the southern core, and gradually spread to deeper waters and to the northernmost core location. This pattern



indicates a gradual expansion of the OMZ during the last deglaciation, as a result of increasing surface productivity in the Eastern Equatorial Pacific and decreasing advective oxygen supply to intermediate waters off Peru.

## 1 Introduction

Oxygen Minimum Zones (OMZs) occur where intense upwelling and high primary productivity result in elevated oxygen consumption within the water column in combination with sluggish ventilation (Wyrtki, 1962; Helly and Levin, 2004; Fuenzalida et al., 2009). In today's world oceans, the most pronounced OMZs with oxygen concentrations <20 µmol/kg are observed offshore northwest and southwest Africa, in the Arabian Sea and Bay of Bengal in the Indian Ocean, and along the continental margin of the Eastern Pacific at low latitudes (Helly and

Levin, 2004; Paulmier and Ruiz-Pino, 2009). Warmer conditions have contributed to expansion of OMZs during the last decades (Stramma et al., 2008; Schmidtko et al., 2017; Levin, 2018; Oschlies et al., 2018). Paleoceanographic reconstructions of bottom-water oxygenation during the last deglaciation are a valuable approach to understand the dynamics of OMZs during changing climatic conditions (e.g., Jaccard and Galbraith, 2012; Moffitt et al., 2015; Praetorius et al., 2015). The Eastern Equatorial Pacific (EEP) has been the focus of

various paleoceanographic studies to unravel the dynamics of surface productivity and bottom-water oxygenation (Oberhänsli et al., 1990; Heinze and Wefer, 1992; Cannariato and Kennett, 1999; Loubere, 1999; Hendy and Pedersen, 2006; Martinez and Robinson, 2010; Moffitt et al., 2014; Scholz et al., 2014; Salvatteci et al., 2016; Tetard et al., 2017; Balestra et al., 2018; Hoogakker et al., 2018). The region is characterized by a strong and shallow OMZ maintained as a result of persistent upwelling (Pennington et al., 2006). Previous studies in the

region used established paleo-proxies such as sedimentary textures (laminations), productivity indicators ($C_{org}$, $\delta^{15}N$, biogenic opal), redox sensitive elements (e.g., U, Mo, Cd, V, Mn) and benthic foraminiferal distributions (e.g., Jaccard et al., 2014; Moffitt et al., 2015). The most of the studies reported that cold, glacial periods were generally associated with contracted OMZ, whereas warm or interglacial periods were associated with an expansion of the OMZ at intermediate depths. However, only a few studies attempted bottom-water oxygen

reconstructions of the Peruvian margin, and they used records from sediment cores recovered from depths shallower than 400 m (Oberhänsli et al., 1990; Heinze and Wefer, 1992; Scholz et al., 2014; Moffitt et al., 2015; Salvatteci et al., 2016). In a review, Schönfeld et al. (2015) demonstrated that the deposition of laminated sediments indicated oxygen concentrations <7 µmol/kg, and that accumulation rates of sedimentary organic carbon could be used to quantify oxygen concentrations of >10 µmol/kg. A geochemical approach focusing on

the Peruvian margin used redox sensitive elements (Fe, Mo, U) and found a 5 to 10 µmol/kg decrease from




glacial to interglacial periods in the centre of the OMZ at depths between 100 m and 500 m (Scholz et al., 2014).
Because of the lack of complete records from the continental slope off Peru (Reimers and Suess, 1983; Erdem et
al., 2016) and the limited applicability of some of the redox proxies (laminated sediments, Mo, U) at higher
oxygen levels, paleo-oxygen reconstructions were not possible at the lower, dysoxic - oxic boundary of the

Peruvian OMZ. However, benthic foraminiferal faunas were markedly structured with oxygenation at these
depths (Mallon et al., 2012). Therefore, the present study aimed to reconstruct paleo-oxygen conditions since the
Last Glacial Maximum (LGM) by using benthic foraminiferal records from sediment cores from the Peruvian
OMZ between 600 and 1250 m water depth. We compiled all available information on living (rose Bengal
stained) benthic foraminiferal faunas from the Peruvian margin as calibration dataset to investigate the following

questions: 1) did the Peruvian OMZ structure show differences in terms of vertical and horizontal extension since
the LGM? 2) if there are such differences, can we actually quantify these changes in bottom-water oxygen
concentrations ($[O_2]_{BW}$) by reverting to the oxygen demands of today's living faunas? 3) and finally if so, how
much did the $[O_2]_{BW}$ levels change since the LGM?

## 1.1 Benthic foraminifera as oxygen proxy

Certain benthic foraminiferal species and assemblages have been suggested as proxies for bottom-water
oxygenation, in particular for low-oxic to anoxic conditions (e.g., Sen Gupta and Machain-Castillo, 1993; Kaiho,
1994; Alve and Bernhard, 1995; Bernhard et al., 1997; Baas et al., 1998; Nordberg et al., 2000; Leiter and
Altenbach, 2010). Since a high flux of particulate organic matter to the sea floor prevails in OMZs, these species
also flourish under elevated food availability. Applications of certain species to reconstruct ancient bottom-water

oxygen concentrations were often undermined by the TROX-model (Barmawidjaja et al., 1992; Jorissen et al.,
1995). This conceptual model explains the microhabitat structure of benthic foraminifera in the sediments as
driven by both, the availability of the organic matter and dissolved oxygen (Van der Zwaan, 1999). A growing
number of publications reporting the living (rose Bengal stained) benthic foraminiferal distributions and their
ambient environmental conditions (Phleger and Soutar, 1973; Mackensen and Douglas, 1989; Sen Gupta and

Machain-Castillo, 1993; Bernhard et al., 1997; den Dulk et al., 1998; Jannink et al., 1998; Levin et al., 2002;
Schumacher et al., 2007; Cardich et al., 2012; Mallon et al., 2012; Caulle et al., 2014; Cardich et al., 2015)
showed some features in common: First, benthic foraminiferal faunas generally show a low diversity and high
population density in oxygen-depleted environments; 2) most but not all living specimens in OMZs were found
dwelling in the first one or two cm of the surface sediments here, even at moderate flux rates of particulate

organic matter; 3) species with a thin, porous test wall (e.g., Bolivinids) always outnumber the agglutinated and
porcelaneous species at low oxygen levels. Pore densities of the tests have been recognised as indicators of



bottom and pore-water redox conditions (Kaiho, 1994; Glock et al., 2011; Kuhnt et al., 2013; Rathburn et al., 2018). A comparison of different OMZ settings showed that benthic foraminiferal assemblages and distributions could be used to identify spatial changes of the OMZ provided certain threshold values and ranges are considered (Table 1). Here, we consider the following classification: microxic conditions <5 µmol/kg, dysoxic conditions 5-

45 µmol/kg, oxic conditions >45 µmol/kg. Overall, an extreme low oxygen, even anoxia tolerant association was found within the OMZ core, a transitional species group was recorded around the lower boundary of the OMZ (>20 µmol/kg), and a cosmopolitan and much more diverse fauna was observed outside the OMZ (e.g., Table 2; Schumacher et al., 2007; Mallon et al., 2012; Caulle et al., 2014).

## 1.2 Regional setting

The Peruvian OMZ is one of the most pronounced OMZs in the world (Figure 1; Paulmier and Ruiz-Pino, 2009), covering the Peruvian continental shelf and upper slope, with its thickest part between 5°S and 15°S and 50 to 750 m water depths (Figure 2; Fuenzalida et al., 2009). The intensity of the OMZ is dependent on the low ventilation of advected intermediate waters, diapycnal mixing, and the extremely high primary productivity in the surface waters (Karstensen et al., 2008; Brandt et al., 2015). The productivity is maintained by the wind–driven

upwelling of cold, nutrient-rich, and oxygen-poor waters from intermediate depths (e.g., Pennington et al., 2006). The main source of these upwelled waters is the Peru-Chile Undercurrent (PCUC). It originates around 3-5°S and flows southward between 50 and 300 m water depths (Montes et al., 2010; Chaigneau et al., 2013). The PCUC is fed by the Equatorial Undercurrent (EUC) and Southern Subsurface Countercurrents (SSCCs; Montes et al., 2010). Below the PCUC, northward flowing Chile-Peru Deep Coastal Current (CPDCC) carries cold Antarctic

Intermediate Waters as a thin layer (AAIW; Chaigneau et al., 2013).

## 2 Material and methods

### 2.1 Sediment cores

Four sediment cores were considered in this study (M77/2-47-2; 50-4; 52-2 and 59-1). They were collected during expedition M77 Leg 2 with *R/V Meteor* in 2008 from the continental slope between 3°S and 9°S and water

depths of 600 m and 1250 m around the lower boundary of today's OMZ (Figure 1 and Figure 2, Table 3). The age model of the core M77/2-59-1 (3°57'S 997 m) was established by (Mollier-Vogel et al., 2013). Age models of the other cores were previously described elsewhere (Erdem et al., 2016). Sedimentary $\delta^{15}N$ data of core 52-2 and 59-1 were taken from Glock et al. (2018) and Mollier-Vogel et al. (2019), respectively (Table 3). For this study, we focused on the following time intervals with 300 to 500 year resolution at each core: the late Holocene



(LH;3-5 cal ka BP), the early Holocene (EH; 8-10 cal ka BP); the Bølling Allerød/Antarctic Cold Reversal (BA/ACR; 13-14.5 cal ka BP), the Heinrich Stadial-1 (HS1; 15-17.5 cal ka BP) and the Last Glacial Maximum (LGM; 20-22 cal ka BP). For benthic foraminiferal analyses, 10 to 20 cc sediment samples were wet sieved on a 63 μm screen immediately after they were taken, and the residues were dried at 40°C. They were later split with

an Otto microsplitter when needed, in order to attain similar total numbers of specimens, around 300 per sample (Murray, 2006). The foraminifera were dry picked, collected in Plummer cell slides, sorted by species, fixed with glue and counted. Benthic foraminiferal assemblage compositions and taxonomic references were previously reported elsewhere (Erdem and Schönfeld, 2017).

## 2.2 Surface samples and living benthic foraminifera

Information on the living (rose Bengal stained) benthic foraminifera was compiled from four independent datasets. They comprise 53 samples from the Peruvian continental shelf and slope from water depths of 48 to 2092 m between 1°45'S and 17°28'S (Figure 2, Table 4). Four of the samples from a transect around 12°30'S were collected in December – January 1998 during Panorama Expedition, Leg 3a, with *R/V Melville.* Eight surface sediment samples and environmental parameters were collected from the continental shelf and uppermost

slope around 12°S and 14°S during different monitoring cruises with *R/V SNP 2* and *José Olaya Balandra* in August and April 2009, 2010 and 2011. For the present study, averaged bottom water oxygen values are used for these stations (for details see Cardich et al. (2015)). The largest dataset was gathered from 33 surface samples collected in October to December in 2008 during *R/V Meteor* expeditions M77 Leg 1 & 2 (Mallon, 2012; Mallon et al., 2012). The eight most recent samples and supplementary data were collected in May 2017 during *R/V*

*Meteor* expedition M137 (Sommer, 2017). The surface sediment samples in all studies comprise the topmost 10 or 30 mm. The faunal census is based on the >63 μm size fraction. In case of fractionated subsamples, we combined the values of different grain size fractions considering the volumes and splits reported for each subsample.

### 2.2.1 Data reduction: consideration of taphonomy

The inventory of living benthic foraminiferal species was compared with that from the sediment cores after compilation of the joint dataset. As expected, only single specimens of three agglutinated species were found in some samples from core 50-4 and 52-2. Agglutinated species have a lower preservation potential after burial in the sediment because their organic cement is decomposed during early diagenesis or changing redox conditions (Schröder, 1988; Mackensen et al., 1990). Consequently, only species with calcareous tests were considered for

further analyses. Even though agglutinated species were not used for our downcore application, they are well





known for their low tolerance to oxygen minimum conditions (Bernhard and Bowser, 1999; Gooday and Rathburn, 1999; Gooday et al., 2000; Levin et al., 2002; Mallon, 2012). We therefore considered their abundances separately, and the proportions of agglutinated species in living faunas were used for a comparison of the results.

Another exception was made for *Hoeglundina elegans.* The aragonitic test of this species has a low preservation potential (Gonzales et al., 2017). *Hoeglundina elegans* was observed in four surface samples with more than 5 % abundance but it was almost absent from the foraminiferal assemblages of sediment cores. Therefore, we excluded *H. elegans* from the census data as well. The faunal data from surface sediment and core samples were reduced accordingly.

Considering the water depths at which the sediment cores were taken, (>600 m) and the depth range of our surface samples (Figure 2), we further reduced the dataset by taking only surface samples collected from water depths >300 m into account. Species showing at least three occurrences with percentages of more than 5 % from these samples were listed and considered for further analyses. The final reduced dataset included in total 16 species from 35 samples.

**2.2.2 Environmental data**

Bottom water oxygen concentrations ($[O_2]_{BW}$) were measured at the time of sampling and reported to vary between 0.0 and 100.4 µmol/kg (Table 4). Dissolved oxygen concentrations at the *R/V Melville* stations around 12°30'S were previously reported (Levin et al., 2002). For the eight *R/V SNP 2* and *José Olaya Balandra* stations, $[O_2]_{BW}$ data were measured on each cruise (Cardich et al., 2015). We thus considered mean values. The $[O_2]_{BW}$ of
the *R/V Meteor* M77 stations were taken from Mallon et al. (2012) and from a synoptic compilation of all CTD hydrocasts from this area (Schönfeld et al., 2015). The $[O_2]_{BW}$ of the *R/V Meteor* M137 stations were extracted from the expedition dataset (Gerd Krahmann, GEOMAR, pers. comm.). Rain rates of particulate organic carbon (RRPOC; mmol/m²d) for six of the stations were taken directly from (Dale et al., 2015; Table 4). The rain rates for the other stations were estimated by using equations provided for water depths between 100 and 1000 m
(Martin et al., 1987; Dale et al., 2015). Different primary productivity values were used for the RRPOC calculations. For the region around 11-12°S, values reported in the Supplement of Dale et al. (2015) were used since the measurements were done close to sampling during the M77 expedition Mallon et al. (2012). For the region around and south of 15°S, values reported by Martin et al. (1987) were considered. For the northern part of the study area, estimates from Pennington et al. (2006) were used.





## 2.3 Statistical analyses

We standardized the reduced data matrix by calculating the proportions of the involved species referring to the total calcareous (calcitic) species of each sample as described above. Relative abundances (percentages) were preferred instead of absolute abundances (individuals per cm$^3$) since this information would create an

inconsistency when applied to surface sediment samples and to fossil sediments in the same manner. Diversity and dominance values were calculated for the reduced calcareous (calcitic) assemblages, together with *Q*-mode hierarchical cluster analysis and Canonical Correspondence Analysis (CCA), including comparison with the environmental variables  [O$_2$]$_{BW}$ and RRPOC.

All statistical and diversity analyses were performed with the PAleontological STatistics (PAST) software,

Version 3.11 (Hammer et al., 2001). *Q*-mode hierarchical cluster analysis were applied using the Unweighted Pair Group Method (UPGMA) based on a Bray-Curtis similarity matrix. Canonical Correspondence Analysis (CCA) was performed to the same datasets to see the relation between the species, stations and environmental variables. Additionally, multiple regression analysis was applied and their significance was assessed in order to evaluate their reliability for downcore applications. Coefficients and intercept value from the multiple regression

analysis was later used as data entry for a polynomial transfer function to calculate past bottom-water oxygen concentrations from foraminiferal assemblages of sediment core samples.

## 3 Results

### 3.1 Living benthic foraminiferal distributions

The entire dataset of living calcareous benthic foraminifera from the Peruvian margin comprised 53 surface

sediment samples and 127 different calcareous (calcitic) species. When the oxic (>45 μmol/kg), dysoxic (5-45 μmol/kg), and microxic (<5 μmol/kg) classification was applied to this dataset, 27 samples were classified as microxic, 20 samples were classified as dysoxic and the remaining 6 samples were classified as oxic (Figure 3). The diversity of calcareous species and the relative abundance of the total of agglutinated species increased with bottom-water oxygen. In particular, a marked increase in the proportion of agglutinated species was observed at

stations with [O$_2$]$_{BW}$ >15 μmol/kg, and a further increase was recorded around 40 μmol/kg. The proportion of the agglutinated taxa was more than 50 % at sampling stations under oxic conditions. The assemblages were, however, less diverse as compared to dysoxic samples, as depicted by low Fisher alpha indices and the dominance of single species. This pattern implies that these six oxic stations did not represent the total calcareous taxa very well; neither the stations do in the reduced dataset of 16 species and 35 samples that we statistically

analysed. The microxic samples showed less diverse assemblages with higher dominances, whereas the dysoxic



samples showed higher diversities and a lower dominance. The Fisher alpha diversity indices were lower and dominance was higher at stations under dysoxic conditions where the living faunas were dominated by Bolivinids (Figure 3).

In the centre of the OMZ, at microxic stations, the most abundant species was *Bolivina seminuda* (supplementary figure S.1), followed by *B. costata*, *B. interjuncta*, *B. spissa* and *Bolivinita minuta* with increasing oxygen concentrations. *Uvigerina peregrina* became the most abundant species outside the OMZ core showing similar trends as agglutinated species and other calcareous species which increased markedly as well. *Pyrgo murrhyna* and *Melonis barleeanum* were observed only at oxic stations. *Q*-mode hierarchical cluster analysis on the entire census data (53 samples and 127 calcareous species) indicated three clusters (supplementary figure S.2). The census dataset was later reduced to most abundant 16 species from stations deeper than 300 m. All of the species from this subset were grouped in two clusters (A and B; supplementary figure S.2). Application of CCA on these abundant species showed similar results (Figure 4). Most of the samples and species grouped together as indicated by their positive loading of Axis 2. Overall, lower $[O_2]_{BW}$ and higher RRPOC did not show an eye-catching relation as expected. When multiple regression results were used to compare these observations, the fitted $[O_2]_{BW}$ showed a significant correlation with measured values at the sampling stations whereas the RRPOC estimates did not: $R^2 = 0.82$, $p<0.05$ and $R^2 = 0.53$, $p=0.31$, respectively (Figure 5 and Table 5).

## 3.2 Application of $[O_2]_{BW}$ estimates to sediment cores

Erosion, reworking and high energetic bottom conditions prevail at the continental slope of the Peruvian margin. The Holocene from cores 47-2 and 50-4 were missing (Erdem et al., 2016), and due to the high sedimentation rates, core 59-1 covers only the late Holocene (LH), early Holocene (EH), Bølling Allerød/Antarctic Cold Reversal (BA/ACR), and Heinrich Stadial-1 (HS1). Consequently, it was only possible to compare these time intervals in all of the sediment cores. In the following, we describe the results of $[O_2]_{BW}$ quantification for each core separately, from south to north. We abstained from applying the same approach to reconstruct past RRPOC for the downcore records because the regression analyses did not show a significant correlation. Additionally, three sediment cores were recovered around or deeper than 1000 m which was the maximum depth for any reliable RRPOC calculations.

Core M77/2-47-2 was retrieved from 626 m water depths, and the prevailing oxygen at this location was around 10 μmol/kg during the time of sampling. The sedimentation rates did not exceed 7.5 cm/ka. The time intervals, the BA/ACR, the HS1 and the LGM, were covered by 12 samples in total. Agglutinated species were absent from all samples. The paleo-oxygen estimates showed rather similar values and no profound fluctuations over time (Figure 6). The $[O_2]_{BW}$ ranged from 12 μmol/kg to 28 μmol/kg with lowest values around 15 cal ka BP. The



standard deviations varied between 14 and 19 µmol/kg. Core M77/2-50-4 was collected from 1013 m water depth and the $[O_2]_{BW}$ was 52 µmol/kg at the coring station. In total, 20 samples were analysed covering the time intervals BA/ACR, HS1 and LGM, the Holocene was missing in this core. Among the 138 benthic foraminiferal species identified, only one was agglutinated (*Dorothia goesi*) and occurred as single specimen in samples from

the early HS1 and LGM. The estimated paleo-oxygen concentrations ranged from 35 to 43 µmol/kg during the LGM, varied between 21 and 40 µmol/kg during the HS1 and between 9 and 15 µmol/kg during the BA/ACR. Deviations ranged from 9 to 17 µmol/kg. Core M77/2-52-2 was collected from 1249 m water depth. The $[O_2]_{BW}$ was 74 µmol/kg at the core location during sampling. M77/2-52-2 is the only core which spans all time intervals considered in this study. In total, 27 samples were analysed, and 170 species were identified of which three were

agglutinated. The estimated $[O_2]_{BW}$ indicated stable condition during the LGM with values ranging from 52 to 61 µmol/kg, which was followed by a decrease from 56 to 46 µmol/kg during HS1, and a further much more distinct decrease during the BA/ACR from 53 to 10 µmol/kg. After the deglaciation, the fluctuations were weaker indicating rather stable conditions with values of 13 to 30 µmol/kg during the EH and 23 to 32 µmol/kg during the LH. The standard deviations (1sd) ranged from 8 to 20 µmol/kg. Core M77/2-59-1 was recovered from the

northernmost part of the study area, from 997 m water depths, and the $[O_2]_{BW}$ was 54 µmol/kg during sampling. The core location has been under the influence of strong riverine input. The sedimentation rates throughout the core were 50 and 170 cm/ka, rather high as compared to the other cores (Mollier-Vogel et al., 2013). Because of these high sedimentation rates, the LGM was not retrieved by this core. In total, 20 samples were analysed and 161 species were identified. Agglutinated species were scarce but present. They were more frequent in samples

older than 9 cal ka. The estimated $[O_2]_{BW}$ ranged from 34 to 60 µmol/kg during HS1, from 63 to 18 µmol/kg during the BA/ACR, varied between 11 and 30 µmol/kg during the EH and between 3 and 25 µmol/kg during the LH. One sample showed a negative value, obviously an artefact of the method. Overall standard deviations were calculated as ranging from 9 to 28 µmol/kg. When the average values were considered in each time interval, the decrease from HS1 to the BA/ACR was 16 µmol/kg in core 50-4, 20 µmol/kg at core 52-2 and only 6 µmol/kg in

core 59-1. Whereas the shallowest core 47-2, indicated that $[O_2]_{BW}$ values slightly increased (5 µmol/kg) from HS1 to BA/ACR. The difference between the BA/ACR and the early Holocene $[O_2]_{BW}$ was around 11 µmol/kg at core 52-2 and around 24 µmol/kg at core 59-1.



## 4 Discussion

### 4.1 Living benthic foraminifera in relation with OMZ settings

Similar to previous observations from other modern OMZs, benthic foraminifera living in the Peruvian OMZ showed high population densities and low diversity in the centre. Certain species, predominantly Bolivinids, were observed to be the most abundant species here. They are known for their high-tolerance to suboxic even anoxic conditions (Table 2; Mullins et al., 1985; Schumacher et al., 2007; Piña-Ochoa et al., 2010; Glock et al., 2011; Mallon et al., 2012; Cardich et al., 2015). In the vicinity of the OMZ core, around the boundary, a more diverse assemblage (cluster A; supp. Figure S.2) was observed. Considering the environmental information gathered on some of the species of this group, such as *Bolivina spissa*, *Bolivinita minuta, Cassidulina delicata, Epistominella pacifica* and *Uvigerina peregrina* (Table 2)*,* the assemblage seemingly represented a transitional community which reacted to a broader range of environmental variables including increasing water depths, oxygen content and organic carbon content. The living taxa from stations below the lower boundary of the OMZ indicated much more diverse assemblage (cluster B; supp. Figure S.2). This assemblage involved mostly Miliolids which are known for their intolerance to low-oxygen (e.g., Caulle et al., 2014). Agglutinated species dominated the whole assemblage from these samples which again mirrors rather oxic conditions. The data is in good agreement with previous observations on the low tolerance of agglutinated species to oxygen depleted conditions (<0-20 ml/l = ~9 μmol/kg; Gooday et al., 2000)). In the CCA on the living fauna, this Miliolids associated assemblage (cluster B), together with the agglutinated species, showed a positive relation with $[O_2]_{BW}$ (supp. Figure S.3). Similar trends of such different assemblages in relation with changing oxygen and organic matter were also observed in the Arabian Sea (Jannink et al., 1998; Caulle et al., 2014). Even though there are different species involved in the assemblages representing the OMZ core (microxic), around the boundary of the OMZ (dysoxic) and outside the OMZ (oxic), the transitional appearance of these assemblages from low diversity – high density to more diverse and cosmopolitan assemblages indicate strong similarities.

Comparison of living taxa in different OMZ settings revealed that each OMZ has its own genuine assemblages. The most abundant species were not observed at similar abundances in other OMZs indicating that they are specifically adapted to the conditions in these regions. For example, *Bolivina dilatata* is dominant in the Arabian OMZ (Jannink et al., 1998) and *Bolivina costata* is frequent in the core of the Peruvian OMZ (This study; Cardich et al., 2012; Mallon et al., 2012). While *Bolivina dilatata* is widely distributed in the Atlantic Ocean too, including the Mediterranean, *Bolivina costata* is seemingly endemic to the western South American margin. This "trapped" occurrence might be due to the structure and shape of the OMZ, prevailing since more than 0.5 million of years, and the strong adaptation of these assemblages to conditions during a long time (e.g., Heinze and Wefer,




1992). Similar assumptions were made for the OMZ core assemblages in the Northern Arabian Sea (Jannink et al., 1998). Therefore, expecting to find the same specific species within the same oxygen range in different OMZs is potentially misleading. Although there are not many frequent species observed in different OMZs, *Bolivina seminuda* and *Bulimina exilis* are within the few common species (Table 2; Bernhard et al., 1997; den Dulk et al., 1998; Cardich et al., 2015). Elevated proportions of these two species could be used as extreme-low bottom-water oxygen indicator in downcore records (e.g., McKay et al., 2015; Praetorius et al., 2015; Tetard et al., 2017). Additionally, total numbers of the Miliolids could be oxic condition indicators as previously suggested (den Dulk et al., 2000) but this approach would not produce sensible results outside OMZ settings. Accordingly, comparisons of relative abundances of these low-oxygen tolerant and intolerant species in the fossil record might be used in determining dysoxic-oxic transitions (e.g., Kaiho, 1994; Cannariato et al., 1999; Schmiedl et al., 2003; Tetard et al., 2017; Balestra et al., 2018).

### 4.2 Peruvian margin oxygen history since the LGM

The records revealed a distinct decrease in oxygen during deglaciation in all of the cores from the lower OMZ boundary. Three cores did not cover all the considered time intervals, limiting the spatial delineation of oxygenation changes between the LGM and the Holocene. When the records were stacked, the estimates showed a decreasing trend starting from the LGM, a distinct drop during the deglaciation with fluctuations from the HS1 to the BA/ACR, and a slight increase followed by relatively stable concentrations during the Holocene. None-the-less, Holocene $[O_2]_{BW}$ were still lower than those of the LGM. This trend is consistent with other results reported in reviews of bottom-water deoxygenation during the deglaciation in the Eastern Pacific Ocean (Jaccard and Galbraith, 2012; Moffitt et al., 2015). Both reviews consider different proxies (e.g., lamination, $\delta^{15}N$, redox sensitive elements) which are known to indicate oxygen depleted conditions as recorded by sediment cores from above 500 m depths off Peru. Comparing the differences between our cores, the overall decrease was greatest in the deepest core, implying that the absolute changes at the lower boundary were larger than in the core of the OMZ. This might be a reason why profound oxygenation changes were not recorded in core 47-2. Another quantification approach compared Fe concentrations and Mo/U ratios in core M77/2-24-5 from the upper slope off Peru at 11°S and 210 m water depths (Scholz et al., 2014). They found a drop of 5 to 10 µmol/l during the deglaciation, which corroborated our estimates. Furthermore, differences and delays were observed in the timing of the decreases among the different cores. In the southern core 50-4, the most distinct decrease started with the onset of the HS1, whereas the decreasing trend did not commence before the end of this period in the slightly deeper core 52-2 (Figure 6). The amount of the oxygen decrease from HS1 to the BA/ACR was around 15 to 25 µmol/kg in the cores 50-4 and 52-2, whereas it was only 5 µmol/kg in the northern core. At this core, the values



raised later by 11 µmol/kg in the BA/ACR to the Holocene transition whereas the decrease was only around 6 µmol/kg in the core 52-2. This indicated a south to north time-transgressive and decreasing trend in $[O_2]_{BW}$. The downcore distribution of benthic foraminiferal species that were not included in the quantification approach, provided further, corroborating evidence. For instance, the disappearance of *Prygo murrhyna* and agglutinated species after HS1 at core 50-4, and the increasing abundances of *B. costata* during the same time interval suggested that bottom-water conditions successively became dysoxic (Erdem and Schönfeld, 2017). Similarly, the presence of *P. murrhyna* and agglutinated species throughout core 52-2 suggested that the prevailing oxygen levels have been moderate during the considered time intervals. Due to hiatus and sampling resolution our assessments for the Holocene is limited. Besides, large standard deviations observed in the northernmost core 59-1 raised questions about the applicability of the method for the Holocene (Figure 6). Never-the-less; both Holocene records of core 59-1 and 52-2, suggested a slight recovery of the OMZ that is in accordance with recently published results from the region (Salvatteci et al., 2016; Salvatteci et al., 2018; Mollier-Vogel et al., 2019). Since the records are not continuous, we cannot constrain changes in $[O_2]_{BW}$ after the late Holocene. Moreover, we are confident in the $[O_2]_{BW}$ differences in each time interval considered, even though the absolute estimates for each sample might be biased because of the dominance of the low-oxygen samples in the reference dataset.

## 4.3 Comparison with other proxies and records from the region

Total organic carbon content (TOC (w.%)) from core 52-2 and 59-1 showed an increasing trend with the onset of the deglaciation followed by relatively similar trends during the Holocene (Doering et al., 2016; Mollier-Vogel et al., 2019). Higher TOC values during the Holocene suggested higher preservation and thus, enhanced productivity and relatively low oxygen values prevailing at core locations since the EH in comparison to the LGM. During the deglaciation, TOC (w.%) values indicated opposite fluctuations suggesting different environmental conditions prevailing at core locations, i.e., changes in intermediate depth ventilation, riverine input around core 59-1 (Mollier-Vogel et al., 2013). Meanwhile, sedimentary $\delta^{15}N$ values from both cores indicated a higher surface productivity and/or water column denitrification, hence a stronger OMZ during the deglaciation (Figure 7; Glock et al., 2018; Mollier-Vogel et al., 2019). At core 59-1, the increase started not earlier than at the onset of the BA/ACR, whereas core 52-2 showed an increase at the onset of the HS1 already. These results are in agreement with our oxygen reconstructions for the same cores. In core 52-2 for instance, the deoxygenation at the end of HS1 followed relatively stable conditions while $\delta^{15}N_{sed}$ showed lower values and increased again not earlier than at the beginning of the BA/ACR. This coupling of $\delta^{15}N_{sed}$ and bottom-water oxygen suggested a recovery of oxygen depletion in the lower boundary of the OMZ, which is in accordance with





the pore-water nitrate reconstruction from the same core (Glock et al., 2018). Later in the record both cores indicated a distinct drop in the bottom-water oxygen concentrations that were also mirrored by the increasing denitrification during the BA/ACR as displayed by high $\delta^{15}N_{sed}$ values.

Other redox and productivity proxies from the region revealed similar trends. Laminated sediments indicating

$[O_2]_{BW}$ of <7 μmol/kg (e.g., Schönfeld et al., 2015) were observed widest in extent during HS1, in particular on the continental shelf and upper slope between 10 and 18°S (Erdem et al., 2016). Stacked records from the Peruvian margin at 14°S revealed a correlation between enhanced productivity and low bottom-water oxygen as indicated by increasing Mo/U values within the laminated sediments during the HS1 period (Salvatteci et al., 2016). Enhanced surface productivity, increasing denitrification and bottom water deoxygenation during the early

deglaciation was observed in various records from the Eastern Equatorial Pacific (Pedersen, 1983; Pedersen et al., 1988; Hendy and Pedersen, 2006; Martinez and Robinson, 2010; Bova et al., 2018). The distinct increase in relative abundances of the phytodetritus, bloom-feeding *Epistominella exigua* in core 52-2 during the HS1 and in core 59-1 during the entire deglaciation supported these observations indicating an enhanced organic matter flux to the sea floor (Erdem and Schönfeld, 2017). Never-the-less, the highest percentages of *E. exigua* at core 52-2

were recorded when the bottom-water oxygenation indicated more or less stable conditions. This decoupling of productivity and deoxygenation suggested that bottom-water deoxygenation did not always co-occur with enhanced surface productivity at the core location. The discrepancy might be a corroborating evidence that other parameters than primary productivity were influencing the oxygen dynamics in the Peruvian OMZ as they do today, for instance ventilation of intermediate waters or changes in the hydrodynamics (Karstensen et al., 2008;

Brandt et al., 2015). However, it should be kept in mind the Peruvian margin potentially represents a much more complex structure in terms of productivity as it is mirrored in biogenic opal and $\delta^{30}Si$ records (Doering et al., 2016) as well as impact of different intermediate water masses and stratification (Bova et al., 2015; Bova et al., 2018). The same caution should be applied to our *E. exigua* records. The species was observed in the northern cores but not in core 50-4 situated much closer to the main upwelling area. This might be explained by the upper

food flux tolerance limit of ca. 200 g C m$^{-2}$ yr$^{-1}$ for this species (Altenbach et al., 1999: their Figure 6). Although the productivity signatures in the sediments showed variations among the different cores, the $[O_2]_{BW}$ estimates always showed similar trends. This might be indication of different dynamics in the surface waters than in intermediate waters which need further and broader investigation focusing on different records and proxies. Even though our estimated $[O_2]_{BW}$ values were lower than anticipated, the quantification approach is consistent in

terms of absolute changes and coherent with other proxies. Our data and comparisons with other proxies indicated an expansion of the northern part of the Peruvian OMZ in terms of thickness and wideness during the last deglaciation with a decrease in $[O_2]_{BW}$ of 30 μmol/kg at its lower boundary.



## 5 Conclusions

The use of benthic foraminiferal assemblages as a bottom water oxygenation proxy has been under debate since the oxygen-deficiency indicator species can be found in many other environments as well. When certain thresholds are applied, for instance microxic (<5 μmol/kg), dysoxic (5-45 μmol/kg), oxic (>45 μmol/kg), benthic foraminiferal assemblages were observed indicating similar transitional trends in different oxygen minimum zone settings world-wide, even though they are composed of different species. The present study reports an extensive dataset based on four independent studies of living (rose Bengal stained) benthic foraminiferal distributions from the continental shelf and slope off Peru. The faunal distribution data were compared with bottom-water oxygen concentrations ($[O_2]_{BW}$) measured during the sampling periods. Certain species and assemblages showed a much better correlation with $[O_2]_{BW}$ than with rain rates of particulate organic carbon (RRPOC). Application of a multiple regression analysis with $[O_2]_{BW}$ as dependant variable indicated that the foraminiferal assemblages along the Peruvian margin are rather governed by oxygen availability than by the deposition of particulate organic matter. The correlation with $[O_2]_{BW}$ was significant ($R^2 = 0.82$; $p<0.05$), therefore we applied the transfer function to four sediment cores taken from the lower boundary of the Peruvian Oxygen Minimum Zone (OMZ) in order to quantify the past $[O_2]_{BW}$. The data revealed a drop in $[O_2]_{BW}$ of 30 μmol/kg at the lower boundary of the OMZ during the last deglaciation. The decrease was largest at the deepest core site whereas differences were not significant closer to the centre of the OMZ. The overall deoxygenation trend started with the onset of Heinrich Stadial 1 and it was first observed at the southernmost core. It was later followed by a distinct drop during the deglaciation in all other cores. A slight increase was observed in the northern cores during the Holocene. This general trend is in line with previous paleo-oxygenation proxy records at intermediate depths from the Eastern Pacific Ocean, supporting the viability of the benthic foraminiferal approach.





**Acknowledgement**

We would like to thank the crew and scientists aboard *R/V Meteor* during the cruises M77 legs 1 and 2 in 2008 and M137 in 2017, *R/V Melville* during Panorama Expedition Leg 3A in 1998. We also thank Wendy A. Cover for her assistance in the laboratory, Renato Salvatecci and Kristin Doering for collaboration and helpful

discussions. This research was supported by the University of California, NSF Grant 98-03861 to Lisa A Levin and AER, the FONDAP-Humboldt Program and a Basque Country Government Fellowship to MEP and it was funded by Deutsche Forschungsgemeinschaft (DFG) through SFB 754 "Climate–Biogeochemistry Interactions in the Tropical Ocean".



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





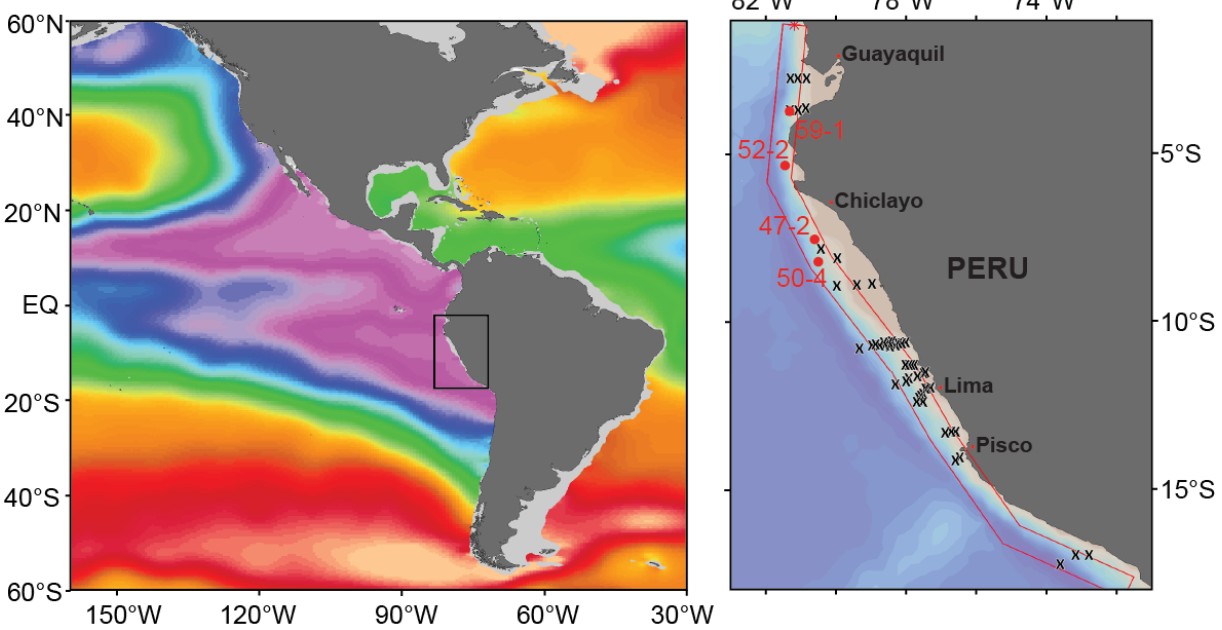

**Figure 1: Location map of the study area (square in panel a) in the Eastern Equatorial Pacific and structure of the OMZ at 400 m water depths. The purple area indicates dissolved oxygen values <0.5 ml/l to <20 µmol/kg; World**
5  **Ocean Database 2013 (Boyer et al., 2013)). b: detail map of the study area showing the locations of the surface samples (x), the sediment cores (red circles). See Figure 2 for the sample locations in relation with water depth and the OMZ intensity.**

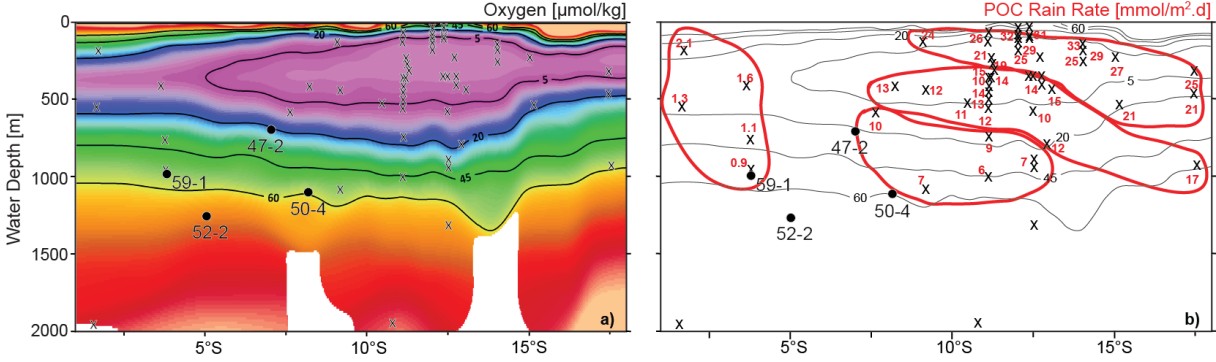

10  **Figure 2: a) Depth vs latitude profile of the dissolved oxygen concentrations measured during the M77 Leg & 2 expeditions (red line in Figure 1, b show the position of this profile; CTD data compilation after Schönfeld et al., 2015); together with the location of the surface samples (x), the sediment cores (circles and core names), b) particulate organic carbon rain rates (RRPOC) calculated for each surface sample (Dale et al., 2015). Rain rates are grouped indicating different values (<5, 5-10, 10-20 and >20 mmol/m²d), see text for details. Contour lines are the same as in a.**





**Figure 3: Relative abundances of Bolivinids and agglutinated species at each station in relation with the bottom-water oxygen measured during sampling. Oxic (>45 µmol/kg), dysoxic (5-45 µmol/kg), and microxic (<5 µmol/kg) classification was applied to these samples and colours green, orange and black were used respectively as indicators of the thresholds. Dominance and Fisher alpha diversity indices were also calculated at the same samples.**



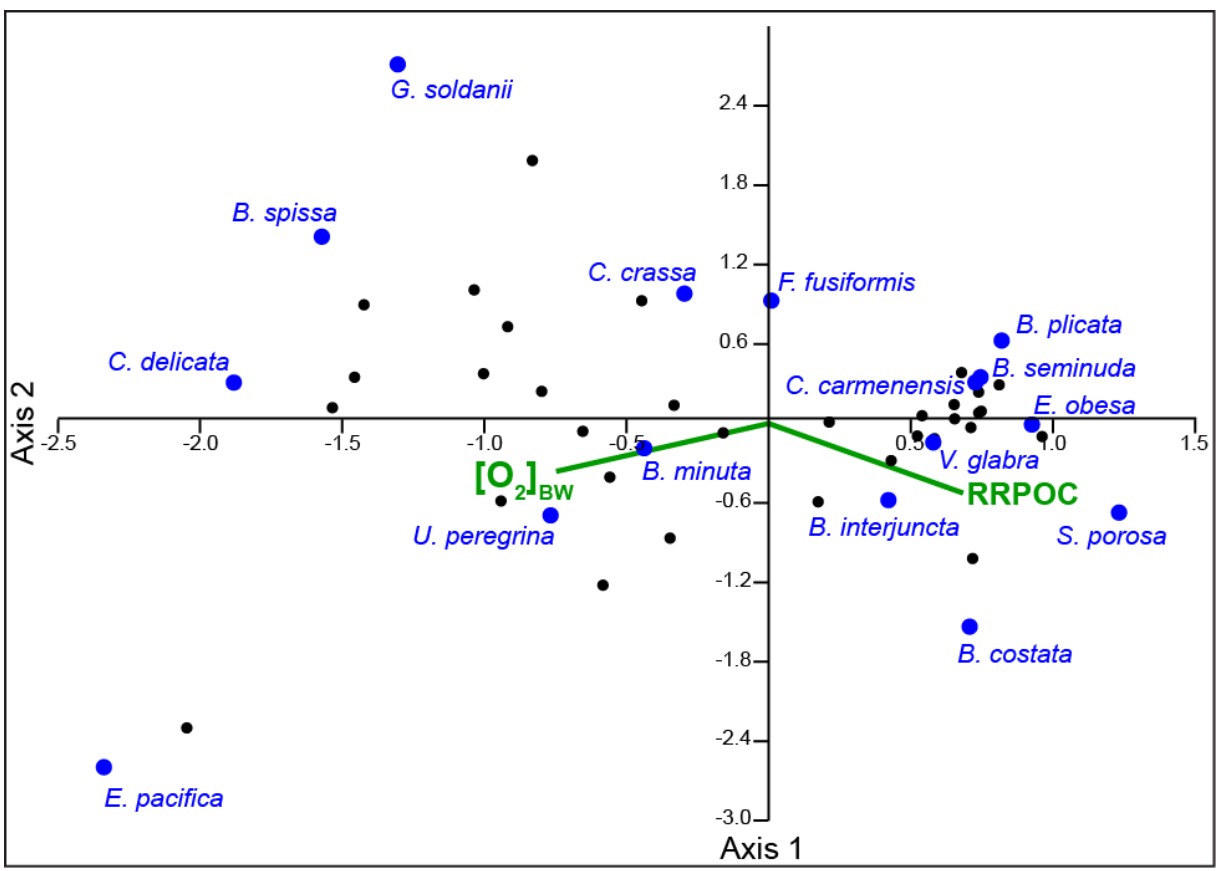

**Figure 4: The results of the CCA application on the most abundant 16 species considered in this study. Black dots are the samples (35 in total) and blue dots are the species involved in the statistical analyses. See Table 5 for full names.**





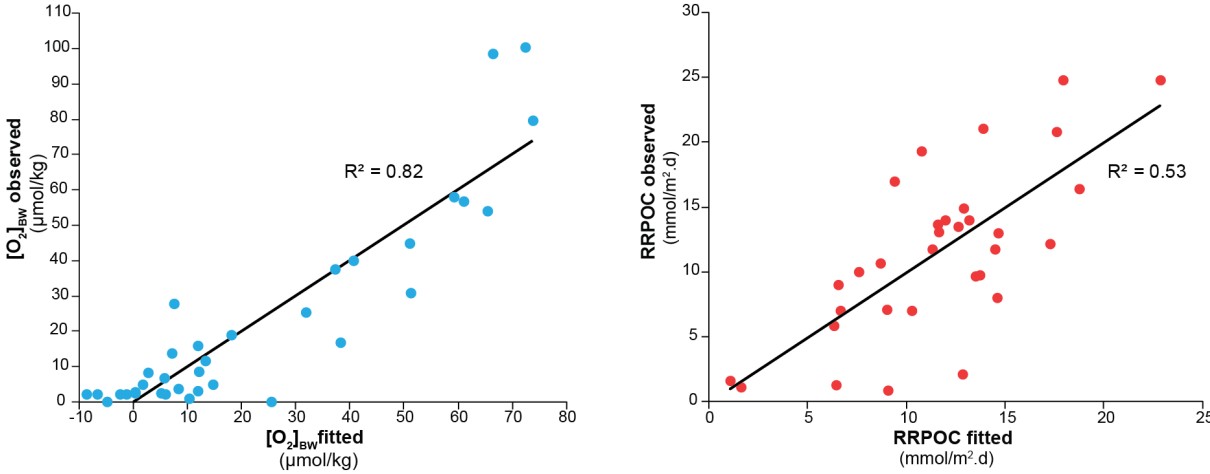

**Figure 5: Graphs showing the regression results applied to the living benthic foraminifera dataset for bottom water oxygen and rain rates.**





**Figure 6: Comparison of the estimated bottom-water oxygen concentrations [O₂]ᴮᵂ applied to four sediment cores from south to north and shallow to deep regarding to their location on the continental slope and the OMZ. Stars indicate the modern [O₂]ᴮᵂ measured during sampling in 2008.**





**Figure 7: Comparison of [O₂]ʙᴡ estimated concentrations with δ¹⁵N and TOC (w.%) from the same cores (data is available for northern cores 59-1 and 52-2).**





**Table 1. Classification of different environments and thresholds of bottom-water dissolved oxygen ($[O_2]_{BW}$) for benthic foraminifera and benthic biota in general**

| Reference | $[O_2]_{BW}$ Range (ml/l) | $[O_2]_{BW}$ Range (µmol/kg) | Classification |
|---|---|---|---|
| Tyson and Pearson (1991) | 2-8 | 90-360 | oxic |
| Sen Gupta and Machain-Castillo (1993) | 0.2-2 | 9-90 | dysoxic |
| | 0-0.2 | 0-9 | suboxic |
| | 0 | 0 | anoxic |
| Kaiho (1994) | >1.5 | >67.5 | oxic |
| Baas et al. (1998) | 0.3-1.5 | 13.5-67.5 | suboxic |
| Cannariato et al. (1999) | 0.1-0.3 | 4.5-13.5 | dysoxic |
| | <0.1 | <4.5 | anoxic |
| Bernhard and Sen Gupta (1999) | >1 | >45 | oxic |
| Levin (2003) | 0.1-1 | 5-45 | dysoxic |
| Mallon et al. (2012) | <0.1 | <5 | microxic |
| Caulle et al. (2014) | 0 | 0 | anoxic |
| Rathburn et al. (2018) | >2 | >89 | oxic |
| | 0.5-2 | 22-89 | dysoxic |
| | <0.5 | <22 | suboxic |



**Table 2. Common species considered in this study and reported from different oxygen depleted environments.**

| Species | Ecology |
| --- | --- |
| *Bolivina costata* | Characteristic species of the Peruvian margin particularly on the shelf (Resig, 1981, 1990; Cardich et al., 2012); tolerant to low bottom-water oxygen <5 μmol/kg (Khusid, 1974; Resig, 1981; Mallon et al., 2012) even to sulfidic pore-waters (Cardich et al., 2015); used as a proxy for enhanced upwelling during interglacial periods (Heinze and Wefer, 1992). |
| *Bolivina seminuda* | Dominant in different OMZs (Phleger and Soutar, 1973; Hermelin and Shimmield, 1990; Bernhard et al., 1997; Ohga and Kitazato, 1997; Gooday et al., 2000; Cardich et al., 2012; Caulle et al., 2014); tolerant to extreme low oxygen <2.5 μmol/kg (Mallon, 2012; Cardich et al., 2015); able to use nitrate for respiration (Piña-Ochoa et al., 2010); better adapted to low oxygen levels compared to other *Bolivina* species e.g., *B. spissa* (Glock et al., 2011); suggested as dysoxic indicator (Kaiho, 1994), used as a proxy for enhanced upwelling during interglacial periods (Heinze and Wefer, 1992) and for dysoxic conditions (Tetard et al., 2017). |
| *Bolivina spissa* | Common in the OMZs in the Pacific Ocean (Douglas and Heitman, 1979; Ingle et al., 1980; Mackensen and Douglas, 1989; Nomaki et al., 2006; Glud et al., 2009; Fontanier et al., 2014; Venturelli et al., 2018); indicates intermediate hypoxic conditions/lower boundary of the OMZ core (Mullins et al., 1985; Mallon, 2012), potentially able to use nitrate for respiration (Glock et al., 2011): suggested as dysoxic indicator (Kaiho, 1994); used as a proxy for suboxic conditions (Cannariato and Kennett, 1999; Tetard et al., 2017), for intermediate hypoxia (Moffitt et al., 2014), and for bottom water nitrate reconstruction (Glock et al., 2018). |
| *Bolivinita minuta* | Common in the OMZ of the Gulf of Panama (Golik and Phleger, 1977); mostly at the lower boundary/outside the core of the OMZ offshore Peru-Chile (Ingle et al., 1980; Mallon, 2012). Other *Bolivinita* species are associated with sustained organic matter flux (Sarkar and Gupta, 2014; and the references therein). |
| *Bulimina exilis* | Dominant in different OMZs (Smith, 1964; Douglas and Heitman, 1979; |



| | Bernhard et al., 1997; den Dulk et al., 1998; Jannink et al., 1998; Caulle et al., 2014; Cardich et al., 2015); associated with fresh organic matter input (Caralp, 1989); suggested as dysoxic indicator (Kaiho, 1994); used as a proxy for dysoxic conditions (Cannariato and Kennett, 1999; Tetard et al., 2017) and severe hypoxia (McKay et al., 2015; Praetorius et al., 2015). |
|---|---|
| *Cassidulina delicata* | Common in the Eastern Pacific OMZs at the lower continental slope (Uchio, 1960; Ingle et al., 1980), and under intermediate bottom-water oxygen concentrations, lowest observed is 4.5 µmol/kg (Golik and Phleger, 1977; Douglas and Heitman, 1979; Resig, 1981; Mackensen and Douglas, 1989; Kaiho, 1994), used as a proxy for dysoxic conditions (Tetard et al., 2017). |
| *Epistominella exigua* | Cosmopolitan, typical in the deep sea environment, opportunistic, associated with pulsed supply of phytodetritus (Gooday, 1988; Gooday, 1993; Smart et al., 1994) and elevated bottom-water oxygen concentrations (Schmiedl et al., 1997; Gupta and Thomas, 2003; Jannink et al., 1998); reported as one of the dominant species along the Peru-Chile margin (Ingle et al., 1980; Resig, 1981). |
| *Epistominella pacifica* | Dominant offshore Peru and California at dysoxic and suboxic conditions (Khusid, 1974; Douglas and Heitman, 1979; Mackensen and Douglas, 1989); suboxic to oxic conditions in the Gulf of Panama (Golik and Phleger, 1977). |
| *Hoeglundina elegans* | Reported as common in areas with variable organic matter input and elevated oxygen concentrations (Gooday, 2003); and the references therein; (Sarkar and Gupta, 2014; Venturelli et al., 2018); also observed in dysoxic sediments (lowest oxygen values measured 9 µmol/kg; Douglas and Heitman, 1979; Mackensen and Douglas, 1989); indicator of elevated oxygenation (Schmiedl et al., 1997; Geslin et al., 2004). Aragonitic shell, prone to dissolution (Gonzales et al., 2017). |
| *Pyrgo murrhyna* | Suggested as oxic indicator (Kaiho, 1994); associated with low to moderate flux of organic matter and moderate bottom-water oxygen concentrations (Gooday, 2003; Sarkar and Gupta, 2014). Generally, large *Miliolids* are |





| | |
|---|---|
| | reported being restricted to higher oxygen concentrations (in Arabian Sea >16 µmol/kg; (Caulle et al., 2014) and suggested as a proxy for rapid ventilation of oxygen-depleted environments (den Dulk et al., 2000). |
| *Uvigerina peregrina* | Cosmopolitan (Gooday and Jorissen, 2012), associated with high organic matter input (Altenbach et al., 1999; Schönfeld and Altenbach, 2005); common in dysoxic sediments of different OMZs (Smith, 1964; Ingle et al., 1980; Ohga and Kitazato, 1997; Venturelli et al., 2018), particularly outside the OMZ core and at the OMZ lower boundary (Jannink et al., 1998; Mallon, 2012), used as a proxy for suboxic conditions (Cannariato and Kennett, 1999; Tetard et al., 2017). |



**Table 3. Metadata of sediment cores used in this study**

| Cruise | Core name | Year | Lat (S) | Long (W) | Water depth (m) | Age model | TOC (w%) | $\delta^{15}N_{sed}$ (‰) |
|---|---|---|---|---|---|---|---|---|
| **M77/2** | 059-PC1 | 2008 | 03°57.01' | 81°19.23' | 997 | Mollier-Vogel et al. (2013) | Mollier-Vogel et al. (2019) | Mollier-Vogel et al. (2019) |
| **M77/2** | 052-PC2 | 2008 | 05°29.01' | 81°27.00' | 1249 | Erdem et al. (2016) | Doering et al. (2016) | Glock et al. (2018) |
| **M77/2** | 047-PC2 | 2008 | 07°52.01' | 80°31.36' | 626 | Erdem et al. (2016) | N.A. | N.A. |
| **M77/2** | 050-PC4 | 2008 | 08°01.01' | 80°30.10' | 1013 | Erdem et al. (2016) | N.A. | N.A. |



**Table 4. Metadata of surface samples from the living benthic foraminifera dataset. Particulate organic carbon rain rates (RRPOC) of stations indicated in bold were taken from Dale et al. (2015), others were calculated (see text for details). Bottom-water oxygen concentrations were taken from reference publications.**

| Sample | Lat - Long | water depth (m) | $[O_2]_{BW}$ (µmol/kg) | RRPOC (mmol/m².d) | Reference |
|---|---|---|---|---|---|
| Site 305 | 12°22.70' 77°29.10' | 305 | 0.89 | 14 | Levin et al., 2002 |
| Site 562 | 12°32.5' 77°29.6' | 562 | 11.61 | 10 | Levin et al., 2002 |
| Site 830 | 12°32.8' 77°34.8' | 830 | 37.5 | 7 | Levin et al., 2002 |
| Site 1210 | 12°40.3' 77°38.5' | 1210 | 79.46 | | Levin et al., 2002 |
| 540 | 11°00.01' 77°47.41' | 79 | 5.28 | **15.3** | Mallon, 2012 |
| 694 | 9°02.97' 79°26.88' | 115 | 1.96 | 24.2 | Mallon, 2012 |
| 470 | 11°00.00' 77°56.61' | 145 | 3.33 | 25.7 | Mallon, 2012 |
| 772 | 1°57.01' 81°07.23' | 207 | 30.8 | 2.1 | Mallon, 2012 |
| 676 | 11°05.01' 78°00.91' | 211 | 2.03 | 21 | Mallon, 2012 |
| 635 | 15°04.75' 75°44.00' | 214 | 2.37 | 27.6 | Mallon, 2012 |
| 583 | 11°06.86' 78°03.11' | 248 | 2.11 | 19.2 | Mallon, 2012 |
| 582 | 11°09.70' 78°04.93' | 291 | 2.28 | 17.6 | Mallon, 2012 |
| 403 | 17°26.00' 71°51.41' | 298 | 2.5 | 24.8 | Mallon, 2012 |
| 616 | 12°22.69' 77°29.06' | 302 | 2.2 | **8** | Mallon, 2012 |
| 473 | 11°00.03' 78°09.95' | 316 | 2.25 | **9.8** | Mallon, 2012 |
| 449 | 11°00.00' 78°09.97' | 319 | 2.25 | 13.5 | Mallon, 2012 |
| 744 | 3°45.01' 81°07.29' | 350 | 6.8 | 1.6 | Mallon, 2012 |
| 716 | 7°59.99' 80°20.51' | 359 | 2.48 | 13.1 | Mallon, 2012 |
| 482 | 11°00.02' 78°14.17' | 375 | 2.26 | **14** | Mallon, 2012 |
| 692 | 9°17.70' 79°37.11' | 437 | 2.86 | 11.8 | Mallon, 2012 |
| 456 | 11°00.01' 78°19.23' | 465 | 3.79 | 13.7 | Mallon, 2012 |
| 406 | 17°28.01' 71°52.40' | 492 | 25.3 | 21.1 | Mallon, 2012 |
| 516 | 10°59.00' 78°21.00' | 511 | 4.83 | 13 | Mallon, 2012 |
| 553 | 10°26.38' 78°54.7' | 521 | 4.83 | 10.7 | Mallon, 2012 |
| 421 | 15°11.39' 75°34.81' | 522 | 13.83 | 20.8 | Mallon, 2012 |





| 767 | 1°53.49' 81°11.75' | 526 | 19 | 1.3 | Mallon, 2012 |
|---|---|---|---|---|---|
| 487 | 11°00.00' 78°23.17' | 579 | 8.59 | 12.2 | Mallon, 2012 |
| 723 | 7°52.01' 80°31.36' | 627 | 8.23 | 9.7 | Mallon, 2012 |
| 459 | 11°00.02' 78°25.6' | 697 | 15.72 | **9** | Mallon, 2012 |
| 757 | 3°51.01' 81°15.49' | 700 | 40 | 1.1 | Mallon, 2012 |
| 622 | 12°32.74' 77°34.73' | 823 | 27.67 | 7 | Mallon, 2012 |
| 410 | 17°38.40' 71°58.23' | 918 | 58 | 17 | Mallon, 2012 |
| 753 | 3°56.95' 81°19.16' | 995 | 54 | 0.9 | Mallon, 2012 |
| 549 | 10°59.81' 78°31.26' | 1004 | 44.78 | **5.9** | Mallon, 2012 |
| 684 | 9°17.69' 79°53.86' | 1105 | 56.63 | 7.1 | Mallon, 2012 |
| 669 | 10°53.22' 78°46.38' | 1923 | 98.35 | | Mallon, 2012 |
| 776 | 1°45.14' 82°37.47' | 2092 | 100.39 | | Mallon, 2012 |
| C1 | 12°01.90' 77°13.07' | 48 | 10.01 | | Cardich et al., 2015 |
| C2 | 12°02.76' 77°17.27' | 94 | 5.85 | | Cardich et al., 2015 |
| C3 | 12°02.34' 77°22.53' | 117 | 4.8 | 32.4 | Cardich et al., 2015 |
| P1 | 14°01,20' 76°18,78' | 120 | 1.80 | 33.2 | Cardich et al., 2015 |
| C4 | 12°02.93' 77°29.01' | 143 | 6.08 | 29.1 | Cardich et al., 2015 |
| C5 | 12°02.22' 77°39.07' | 175 | 6.08 | 26.1 | Cardich et al., 2015 |
| P2 | 14°04,32' 76°25,20' | 180 | 1.58 | 29.2 | Cardich et al., 2015 |
| P3 | 14°07,50' 76°30,54' | 300 | 2.70 | 24.8 | Cardich et al., 2015 |
| M137 - 681 | 12°13,51' 77°10,77' | 74 | 5.69 | | This study |
| M137 - 641 | 12°16,67' 77°14,99' | 128 | 1.6 | 30 | This study |
| M137 - 695 | 12°16,78' 77°14,98' | 130 | 1.11 | 30 | This study |
| M137 - 608 | 12°23,26' 77°24,28' | 244 | 0.00 | 21 | This study |
| M137 - 776 | 12°24,89' 77°26,29' | 303 | 0.02 | 19.3 | This study |
| M137 - 788 | 12°27,19' 77°29,29' | 413 | 0.00 | 16.4 | This study |
| M137 - 735 | 12°38,14' 77°20,74' | 489 | 2.05 | 14.9 | This study |
| M137 - 670 | 12°31,36' 77°34,99' | 752 | 16.75 | 11.8 | This study |



**Table 5. List of the living benthic foraminifera species which were considered in the transfer function, regression coefficients and 1-sigma errors that were calculated with multiple regression analyses.**

|  | $[O_2]_{BW}$ (µmol/kg]) | | RRPOC (mmol/m$^2$.d) | |
|---|---|---|---|---|
|  | **Coeff.** | **1σ** | **Coeff.** | **1σ** |
| **Constant** | 73.83 | 7.249 | 9.6117 | 2.6353 |
| *Bolivina costata* | -1.011 | 0.34144 | 0.26655 | 0.12413 |
| *Bolivina interjuncta* | -1.0887 | 0.26074 | 0.092808 | 0.094787 |
| *Bolivina plicata* | -1.4618 | 0.89245 | -0.12367 | 0.32443 |
| *Bolivina seminuda* | -0.70357 | 0.1895 | 0.018734 | 0.068888 |
| *Bolivina spissa* | -0.43921 | 0.24046 | -0.1654 | 0.087413 |
| *Bolivinita minuta* | -2.4131 | 0.91546 | -0.10672 | 0.3328 |
| *Cancris carmenensis* | -0.85979 | 0.4664 | 0.099572 | 0.16955 |
| *Cassidulina crassa* | -1.8999 | 1.1847 | 0.17166 | 0.43069 |
| *Cassidulina delicata* | -0.80009 | 0.45983 | -0.13439 | 0.16716 |
| *Epistominella obesa* | -0.4445 | 0.95982 | 0.074865 | 0.34893 |
| *Epistominella pacifica* | -0.1219 | 0.34751 | -0.02116 | 0.12633 |
| *Fursenkoina fusiformis* | -1.0253 | 1.7209 | -0.74266 | 0.6256 |
| *Gyroidina soldanii* | -1.7552 | 0.54323 | -0.27674 | 0.19748 |
| *Suggrunda porosa* | -1.7787 | 1.1695 | 0.34883 | 0.42516 |
| *Uvigerina peregrina* | -0.00054 | 0.561 | 0.2497 | 0.20394 |
| *Valvulineria glabra* | 0.2821 | 1.4686 | 0.29553 | 0.5339 |

|  | $R^2$ | p |
|---|---|---|
| **$[O_2]_{BW}$ (µmol/kg)** | 0.824 | 0.00056 |
| **RRPOC (mmol/m$^2$.d)** | 0.5293 | 0.3131 |