# Peer review of "Bottom-water deoxygenation at the Peruvian Margin during the last deglaciation recorded by benthic foraminifera"

_Biogeosciences, 2019_

## Referee Comment (RC1) · Anonymous Referee #2 · 28 May 2019

Erdem et al use benthic foraminiferal assemblages of life benthic foraminifera as a proxy assess bottom-water oxygen concentrations on fossil benthic foraminifera across the upper Peruvian Margin since the last deglaciation.

I do think this is an interesting study, however there are several important issues that need to be addressed to improve the study and interpretations:

1. In the current format, the authors have not demonstrated that the live population are identical too the dead population in the core tops, and without this evidence down-core reconstructions are not scientifically scrutinized.

2. Information concerning age models of the different cores is missing. The age model

needs fully discussed and shown in the article as it is crucial to consider the context and interpretations of the reconstructions.

3. The authors should have a good look at their data and critically reflect whether their conclusions really reflect the data. The main Figure 6, I presume, shows reconstructed O2 plus error. Main changes seem to occur during deglaciation. There does not appear to be any differences between LGM and core tops/late Holocene (the authors suggest a 30 uM change from the LGM to Holocene at the lower OMZ boundary): -The first site at 626 m shows (within error!) similar O2 values during the LGM as core top; e.g. no statistically significant increase in LGM oxygenation. -The second core at 1013 m: all reconstructed values are below present day values: no significant increase in LGM oxygenation here. -Third core site at 1249 m: LGM oxygen concentrations are lower compared with core top; so no significant LGM increase in oxygenation here. -Fourth core at 997 m: perhaps H1, early deglacial higher O2 values; but no reconstructions for the LGM.

So none of the cores show that the Peruvian margin, at the water depths investigated, was better oxygenated during the LGM compared to today.

---

## Referee Comment (RC2) · Anonymous Referee #1 · 10 Jun 2019

The text is well written, being relatively easy to read and understand, though there are some minor errors. One technical problem with the MS is the figures, where there appear some mistakes (error bars, lack of scale bars, outlined below). Scientifically the approach of the authors is straight-forward, applying a transfer function of foraminiferal census counts to determine the bottom water oxygen concentration or an indication thereof. The robustness of the transfer function could/should be test via bootstrapping or a similar analysis (this would help to determine what is the 'counting error', as in how much a few percent change in the abundance would change the transfer functions resultant oxygen value). Moreover, the fitted/predicted oxygen values give negative values, below 0 (figure 5 to 7), I am unsure whether this is possible with concentration

values?

The authors have considered sources of error, one such source of error is the preservation potential of certain forms. For clarity though, it might be prudent of the authors to state how the agglutinated were removed from the data (pg. 5 Line 29-30), i.e., is there potential for error through a closed sum effect? If the assemblage is counted to a certain number, or split to gain a certain number of grains, by removing data (which has to be done) does this introduce some bias (when considering low abundant species – removing 3 specimens in a count of 300 means a loss of 1%, the 100% would be then based on 297, it also shifts the percentages for the remaining species which is more problematic for rare species than for dominant species)? In a similar vein how reliant is the transfer function on small changes for rare species – and have the authors considered a transformation of the abundance data to reduce the impact of dominance and rarity (e.g. Log the data)?

Have the authors considered more environmental variables (e.g. temperature, salinity, etc), whilst the approach here is to reconstruct bottom water oxygen concentrations the question is, is this the dominant control on the assemblage composition? This is especially important given how regional the dataset is especially when comparing different time periods. How similar or dissimilar are the various assemblages for each time period? And how similar or dissimilar are they between? It is not that I doubt that oxygen is a dominant variable, instead knowing whether there is some variation in the assemblage due to another variable might help to put the results into better context.

The discussion of the data is lengthy - what I miss is a statistical comparison between d15N, TOC and the O2 prediction of the authors – as in, figure 7 is under used. Here the authors could compare their proxy against the previous proxy values statistically (e.g., simple scatter or regression analysis) rather than descriptively (section 4.3).

Finally, I agree with the other reviewer that the age models should be outlined in this paper somewhere (e.g., diamond symbols with the depth in cm in the figures?), given

Pg. 8 Line 18 (: 'Erosion, reworking and high energetic bottom conditions').

Text comments:

Pg. 6 Line 4 -7: agglutinated forms have a lower preservation potential, could this affect the splitting? Removing species from abundance counts does impact the closed sum; Pg. 6 Line 19: Mean values – would it not be better to consider the mean with std dev. to construct the equation? Pg. 6 Line 20: "from a synoptic compilation" what do the authors mean by synoptic (= general, vs. synoptic data)? Pg. 6 Line 25-29: How different are the 'different primary productivity values' used for the RRPOC? What would the values be if the same equation were used? This can be tested by applying each set of values used. Pg. 10 Line 25 – 29: Have you considered placing the various species into comparable niche occupations? The table is a good reference guide for readers, but it would be interesting whether the different species regionally/globally occupy different niches or similar ones. Pg. 12 Line 14-16: "Moreover, we are confident in the [O2]BW differences in each time interval considered, even though the absolute estimates for each sample might be biased because of the dominance of the low-oxygen samples in the reference dataset." – maybe elaborate why you have confidence despite the absolute estimates being biased? And how does the absolute estimates being biased fit with research question 2 and 3?

Section 2.2.2: what is the sensitivity of the CTD and equipment used for oxygen, is there not some lower limit (5 umol/kg) below which the data is not accurately measured? Or at least the reliability is not the best.

Section header 1.1 Benthic foraminifera as oxygen proxy -> 'as an oxygen proxy' or reword as 'as a proxy for oxygen'?

Figure comments:

Figure 1a: scale bar missing – if the authors (as implied by the caption) are trying to demonstrate the low oxygen values how about a single contour around the purple?

Figure 1 Caption: should it read as two units? "<0.5 ml/l to <20 $\mu$mol/kg" the 'to' implies a 'sliding scale'

Figure 2b: perhaps color the symbols to show the different rain rates?

Figure 3: (capitalise R in relative abundance). (Bottom panel) The last datapoint (sample M77/2-776) is forcing the plot's yaxis to be skewed toward higher fisher alpha values so that the values of the other samples are condensed. Consider, perhaps using a logscale for the yaxis of the Fisher Alpha panel, alternatively the authors could exclude from this plot sample M77/2-776 and with a big red arrow just tell the reader the values of this 'outlier'. (Bottom and Middle panel) I assume the bars are 'errorbars' – some seem to be not symmetrical around the datapoint (possible depending on the statistic used) but more importantly Site 830, 1004P1 the error bars are below the datapoint.

Figure 5 to 7: Is it possible to have a negative value for oxygen concentration?

Figure 5: give a 1:1 line.

―――――――――――――――――――

---

## Author Comment (AC1) · 23 Jul 2019

We appreciate the remarks and suggestions of the reviewer and are grateful for the effort the reviewer has invested. Below we respond to each comment (RC: referee comment; AR: Authors' response) and indicate how we plan to revise the manuscript accordingly.

RC1: In the current format, the authors have not demonstrated that the live population are identical to the dead population in the core tops, and without this evidence down-core reconstructions are not scientifically scrutinized.

AR1: We thank the reviewer for addressing this important point, and we take the opportunity to emphasize again that living faunas and dead assemblages are generally different in species composition, and that standing stocks are not mirrored in the concentrations of empty tests in near-surface sediments in most environments. These fundamental differences are intrinsic. Living faunas represent the conditions during the weeks before sampling, whereas dead assemblages are a product of many generations added over an unconstrainable time period. Therefore, species richness and density of the dead assemblages are generally higher than the respective values of the living fauna. Another biasing factor is the taphonomic processes altering the composition of the dead assemblages through time, in particular during successive burial under the influence of different redox and pH conditions. The fossil assemblage in sediment cores thereby differs markedly from the dead assemblage at the sediment surface. Therefore, an identical composition of living faunas and dead assemblages in the topmost layers of sediment cores is impossible to be found, and this should not be imposed as a prerequisite for downcore applications of foraminiferal proxies. The very reason why benthic foraminifera are proven reliable paleoindicators is because they live in equilibrium with the ecological conditions in their immediate environment. Due to their short generation times of usually less than a year, they respond quickly to changes in the setting of abiotic or biotic environmental factors. Once a foraminifera reproduces, an empty test is conveyed to the sedimentary record. If the species is reduced in abundance due to environmental changes, a lower number of empty tests is produced per unit of generation time. Conversely, a species benefiting from the change and increasing in abundance will deliver more tests to the fossil record. A transfer function, as it has been applied in the present study, relates the relative abundances of those species to the change in environmental conditions and accommodates for taphonomic alterations. The reviewer is referred to the interesting textbook of Fisher and Wefer on "Use of Proxies in Paleoceanography" for further reading. In summary: our approach is scientifically valid and verifiable through comparison with other proxies for past oxygen conditions.

RC2: Information concerning age models of the different cores is missing. The age model needs fully discussed and shown in the article as it is crucial to consider the context and interpretations of the reconstructions.

AR2: We follow this suggestion and give a detailed age model description in the revised version. We will modify Figure 6 and add age model tie points for each core. We will also link available Pangaea datasets concerning the radiocarbon dating results of these cores.

RC3: The authors should have a good look at their data and critically reflect whether their conclusions really reflect the data. The main Figure 6, I presume, shows reconstructed O2 plus error. Main changes seem to occur during deglaciation. There does not appear to be any differences between LGM and core tops/late Holocene (the authors suggest a 30 uM change from the LGM to Holocene at the lower OMZ boundary): -The first site at 626 m shows (within error!) similar O2 values during the LGM as core top; e.g. no statistically significant increase in LGM oxygenation. -The second core at 1013 m: all reconstructed values are below present day values: no significant increase in LGM oxygenation here. -Third core site at 1249 m: LGM oxygen concentrations are lower compared with core top; so no significant LGM increase in oxygenation here. –Fourth core at 997 m: perhaps H1, early deglacial higher O2 values; but no reconstructions for the LGM. So none of the cores show that the Peruvian margin, at the water depths investigated, was better oxygenated during the LGM compared to today.

AR3: For modern oxygen values we used the CTD data collected during each expedition at the same time when living benthic foraminifera samples were collected. The stars shown on the figures are indicating the values when the sediment archives were collected. The LGM estimations are indeed either really close or below the actual measurements which is seemingly a concern. We mention our concerns about absolute values and potential bias toward lower oxygen value in quantification in section 4.2. (P. 12, Lines 14-16). Nevertheless, it is possible that bottom waters became more oxic after the late Holocene as reported for the shelf during the last 100-150 years (Cardich

et al., 2019). However, we cannot comment further for the rest of the Holocene trend on the basis of currently available information. As this circumstance is apparently not sufficiently addressed in our Discussion, we will detail the respective paragraphs. Concerning the potential bias towards lower oxygen values, we restrain ourselves making comments such as; 'during the LGM at 1000 m water depth oxygen was 50 $\mu$mol/kg'. We rather focus on the absolute changes between periods and sediment archives. Still we present all the quantification results for each data point in Supplementary information Table 1. The average values for each time period were calculated according to estimations presented here. Unfortunately not all time periods are covered in every core, therefore we emphasize that the results are stacked (P. 11 Lines: 15-17). We are aware that the approximation of 30 $\mu$mol/kg is predominantly influenced by the results of core M77/2-52-2 (LGM ranging between 52 and 61 $\mu$mol/kg vs. late Holocene ranging between 23 and 33 $\mu$mol/kg), since it is the only core which covers all of the concerned periods. Once again we primarily focus on the change in oxygenation rather than reporting absolute values as given facts for these cores. Concerning the results of core M77/2-47-2 from (626 m), which does not indicate any change during the LGM and deglaciation, we conceded that these results are puzzling in the first glance, Nonetheless, this record also shows that when the OMZ intensifies (or diminishes) the change is profound around its borders and the conditions are rather stable close to its centre. This regional dynamics has also been disclosed with other proxy based approaches (as discussed in section 4.2; P.11 Lines: 22-26). Moreover, during the LGM this core location was at least 100 m shallower which was potentially within the OMZ core.

References: Cardich, J., Sifeddine, A., Salvatteci, R., Romero, D., Briceño-Zuluaga, F., Graco, M., Anculle, T., Almeida, C., and Gutiérrez, D.: Multidecadal Changes in Marine Subsurface Oxygenation Off Central Peru During the Last ca. 170 Years, Frontiers in Marine Science, 6, 10.3389/fmars.2019.00270, 2019.

―――――――――――――――――――――――――――

---

## Author Comment (AC2) · 23 Jul 2019

Reply to Anonymous Referee #1:

We appreciate the remarks and suggestions of the reviewer and are grateful for the effort the reviewer has invested. Below we respond to each comment individually and indicate how we plan to revise the manuscript accordingly. For clarity, referee comments are indicated in bold and Authors' comments are indicated in italics.

**The robustness of the transfer function could/should be test via bootstrapping or a similar analysis (this would help to determine what is the 'counting error', as in how much a few percent change in the abundance would change the transfer functions resultant oxygen value).**

*We thank the reviewer for the input, we will try this using the same software (PAST) and implement the outcome in the manuscript where relevant under section 2.3.*

**Moreover, the fitted/predicted oxygen values give negative values, below 0 (figure 5 to 7), I am unsure whether this is possible with concentration values? The authors have considered sources of error, one such source of error is the preservation potential of certain forms.**

*Species observed in living dataset are not identical to fossil assemblages in terms of abundance but similar enough to apply transfer function. Once looked closer, only one data point at one sediment core indicated a negative value (late Holocene at core M77/2-59-1 which is the northernmost core) and the all estimations are positive within the statistical uncertainty. The scale bar of these graphs show negative values since we present the results with the standard deviations and the 1sd at some data points are high, it seems like most of the estimates are below 0. The reference dataset based on living foraminifera is predominantly retrieved from stations from the OMZ centre where oxygen concentrations are really low. Therefore, abundant species considered in the key 16 species are predominantly 'low-oxygen tolerant' ones excluding others that are observed in the downcore record. This potentially results in slight shift in the quantification towards lower values. We will make this more clearly discussed under section 4.2 where relevant.*

**For clarity though, it might be prudent of the authors to state how the agglutinated were removed from the data (pg. 5 Line 29-30), i.e., is there potential for error through a closed sum effect? If the assemblage is counted to a certain number, or split to gain a certain number of grains, by removing data (which has to be done) does this introduce some bias (when considering low abundant species – removing 3 specimens in a count of 300 means a loss of 1%, the 100% would be then based on 297, it also shifts the percentages for the remaining species which is more problematic for rare species than for dominant species)?**

*Comparison of living assemblages with fossil record showed that there is a distinct difference between abundance of agglutinated species. This resulted in large bias in downcore applications and thus we decided to continue our transfer function only with calcareous species. Nevertheless, for the living benthic foraminifera interpretations agglutinated foraminifera are included and these results are presented in supporting information.*

*Moreover, their appearance together with Miliolids at few downcore samples is in accordance with our observations on their intolerance to low-oxygen and more oxygenated bottom waters during the LGM.*

**In a similar vein how reliant is the transfer function on small changes for rare species – and have the authors considered a transformation of the abundance data to reduce the impact of dominance and rarity (e.g. Log the data)?**

*Only common species (at least three occurrences with >5 %) are considered in the reference data set, even though frequent species may also be rare in certain samples. As such, small changes in the proportion of rare species will not affect the results. A log-normal distribution is rather a character of volumetric data (i.e., population densities) rather than percentages. Therefore, a logarithmic transformation of the data was not attempted.*

**Have the authors considered more environmental variables (e.g. temperature, salinity, etc), whilst the approach here is to reconstruct bottom water oxygen concentrations the question is, is this the dominant control on the assemblage composition? This is especially important given how regional the dataset is especially when comparing different time periods.**

*We agree with the reviewer that these are really good on point questions and comments, one should always keep in mind the other factors in such investigations. We present oxygen (and rain rates) in relation with the living benthic foraminifera dataset since these are the only parameters either available or possible to calculate for all these sample locations for living benthic foraminifera dataset. Individual studies mentioned here (few of them are published already (Mallon et al., 2012; Cardich et al., 2015)) discuss the relationship between living benthic foraminifera and environmental factors. For the scope of this study we focused on oxygen concentrations which is potentially a dominant factor in such strong oxygen minimum conditions. It would be indeed an interesting study to combine such an extensive dataset with environmental parameters and statistically test their relationships as Cardich et al. (2015) reported for stations from the shelf. In case of paleo-reconstructions, there are plans for comparison work focusing on single sediment archives with application of different proxies.*

**How similar or dissimilar are the various assemblages for each time period? And how similar or dissimilar are they between? It is not that I doubt that oxygen is a dominant variable, instead knowing whether there is some variation in the assemblage due to another variable might help to put the results into better context.**

*Downcore distribution of benthic foraminifera at the same samples with more emphasize on the taxonomy was already published (Erdem and Schönfeld, 2017). Due to assemblage similarities observed in concerned sediment archives we continued with the transfer function approach. The only sediment core which shows relatively different abundance of certain species is the northernmost core M77/2-59-1 (e.g., Bolivina costata is not as abundant as observed in other sediment cores whereas Bolivinita minuta is more abundant). Obviously, when relative abundances of certain species show distinct changes between periods, that is reflected in the oxygen estimations. For instance B. costata and B. minuta both have a large coefficient (table 5) that has a strong influence on the reconstructed oxygen concentrations. During some periods these species are more abundant indicating lower oxygen levels during these periods.*

**The discussion of the data is lengthy - what I miss is a statistical comparison between d15N, TOC and the O2 prediction of the authors – as in, figure 7 is under used. Here the authors could compare their proxy against the previous proxy values statistically (e.g., simple scatter or regression analysis) rather than descriptively (section 4.3).**

*We thank the reviewer for the suggestion. Any paleo-investigation, proxy based research, needs a comparison with other proxies which are potentially related to conditions aimed to be investigated. Our manuscript aims to investigate paleoxygenation that is relatively difficult with currently available*

*proxies (which are introduced to reader in section 1) with each having certain limitations. By multi-proxy applications we aim to cover these limitations with other proxies. We here focused on only two of these proxies; $\delta^{15}N_{sed}$ and total organic carbon content, since they are available for the same cores. So that we did not extend our investigations to wider region also because it is already a long manuscript as the reviewer pointed out.  Application of statistical tests to see the relation between proxies can be done but might also result in misleading interpretations since each proxy is influenced by each other (particularly oxygen and productivity related ones) and various other environmental factors. Furthermore, a scatter plot will not reveal immediate leads and lags depicting temporal relationships of the proxy records, and its statistical characters will be confined by outliers, which only can be identified and assessed in a meaningful manner if the succession of the parameters versus time is displayed.*

**Finally, I agree with the other reviewer that the age models should be outlined in this paper somewhere (e.g., diamond symbols with the depth in cm in the figures?), given Pg. 8 Line 18 (: 'Erosion, reworking and high energetic bottom conditions').**

*We follow this suggestion and give a detailed age model description in the revised version. We will modify Figure 6 and add age model tie points for each core. We will also link available Pangaea datasets concerning the radiocarbon dating results of these cores.*

**Text comments:**

**Pg. 6 Line 4 -7: agglutinated forms have a lower preservation potential, could this affect the splitting? Removing species from abundance counts does impact the closed sum;**

*We agree with the reviewer that it impact the closed sum.  We still present their relative abundance in the living dataset not to mislead the reader for their occurrence. Hence once statistical tests were applied to only calcareous forms we observe large errors and variations at diversity and dominance measures of some samples (e.g., M77/2-776). However, agglutinated forms are absent in the downcore record which makes transfer function approach not applicable.*

**Pg. 6 Line 19: Mean values – would it not be better to consider the mean with std dev. to construct the equation?**

*The dataset mentioned here are published (Cardich et al., 2015). It concerns revisits the same sampling locations over several years. For this study we used the same datasets but averaged values as if it is one sampling. We realized that we used different wording in different sections (2.2 and 2.2.2) when describing the datasets. We will modify this sentence accordingly.*

**Pg. 6 Line 20: "from a synoptic compilation" what do the authors mean by synoptic (= general, vs. synoptic data)?**

*Schönfeld et al. (2015) presented a compilation of CTD data obtained during cruises R/V Meteor M77 legs 1 to 4 between October 2008 and February 2009.*

**Pg. 6 Line 25-29: How different are the 'different primary productivity values' used for the RRPOC? What would the values be if the same equation were used? This can be tested by applying each set of values used.**

*We only calculated rain rates for sampling locations when the information is not available. Otherwise we used the same equation to calculate the rain rates for most of the sampling locations (Martin et al., 1987). Stations from shallower than 100 m and deeper than 1000 m is not covered unless it was already reported in Dale et al. (2015). Different values mentioned here are the primary production*

*estimates from Pennington et al. (2006) and Martin et al. (1987) concerning latitudinal differences. In our calculations, it is not possible to use the same primary productivity estimations since they were reported showing distinct latitudinal differences (i.e., Equatorial upwelling, 13 mg C m$^{-3}$ d$^{-1}$ vs Peruvian coastal upwelling 145 mg C m$^{-3}$ d$^{-1}$ (Pennington et al., 2006)). We were able to compare the results from Dale et al. (2015) and our calculations for latitudes 11°S to 12°S. Overall results from Martin curve are slightly higher than observations of Dale et al. (2015). Nevertheless, the offset is consistent and it would not impact the observations presented in figure 2b for instance. Meanwhile, we realized a mistake at the figure caption that will be corrected in the revised version.*

**Pg. 10 Line 25 – 29: Have you considered placing the various species into comparable niche occupations? The table is a good reference guide for readers, but it would be interesting whether the different species regionally/globally occupy different niches or similar ones.**

*The aim of the table is to provide an overview of the abundant species observed at the Peruvian margin sediments (both modern and downcore). To some extend we tried to bring out their regional or global occurrences in relation with specific environmental factors. Some species such as Epistominella exigua have relatively high amount of records in publications that can be implemented to certain environmental conditions. However, we should be careful while doing that, especially when there is not much information available, as in case of Bolivina costata. For this reason, we wanted to give a small review on availability of information on these species. We will learn more as the genetic information become available over time.*

**Pg. 12 Line 14-16: "Moreover, we are confident in the [O2]BW differences in each time interval considered, even though the absolute estimates for each sample might be biased because of the dominance of the low oxygen samples in the reference dataset." – maybe elaborate why you have confidence despite the absolute estimates being biased? And how does the absolute estimates being biased fit with research question 2 and 3?**

*This part of the discussion (section 4.2) will be improved accordingly as mentioned earlier under the first referee comment.*

**Section 2.2.2: what is the sensitivity of the CTD and equipment used for oxygen, is there not some lower limit (5 umol/kg) below which the data is not accurately measured? Or at least the reliability is not the best.**

*The oxygen sensor that was used on the CTD was a electrochemical "Seabird" sensor (Clark type) that has a detection limit of 1-2 µmol/kg (Revsbech et al., 2009). In a comparison study to other sensors (STOX vs Clark-type (Clark JR et al., 1953; Revsbech et al., 2009) it has been reported that the actual oxygen concentrations at the Peruvian OMZ can be much less when the seabird sensors reach this limit (up to 2 µmol/kg (Kalvelage et al., 2013). Sometimes it can be at the lower nmol/kg range in this region (Revsbech et al., 2009). The maximum error of the oxygen data is constrained to +/- 0.5 µmol/kg. Therefore, almost zero oxygen conditions are recordable. However, values below 2 µmol/kg should be treated with care when Clark-type Seabird sensor was used. In case of our living dataset this is a concern for three stations out of 53. The oxygen concentrations at these stations were likely even lower according to the Revsbech et al. study. A downward correction of 1 µmol/kg would not influence our transfer function in a statistically significant way.*

**Section header 1.1 Benthic foraminifera as oxygen proxy -> 'as an oxygen proxy' or reword as 'as a proxy for oxygen'?**

*This will be corrected in the revised version*

**Figure comments:**

**Figure 1a: scale bar missing – if the authors (as implied by the caption) are trying to demonstrate the low oxygen values how about a single contour around the purple? Figure 1 Caption: should it read as two units? "<0.5 ml/l to <20 mol/kg" the 'to' implies a 'sliding scale'**

*We will add a contour line to F.1a as suggested and will correct the caption in the revised version.*

**Figure 2b: perhaps color the symbols to show the different rain rates?**

*In the revised version we will group the sampling location with similar rain rates with different colours.*

**Figure 3: (capitalise R in relative abundance). (Bottom panel) The last datapoint (sample M77/2-776) is forcing the plot's yaxis to be skewed toward higher fisher alpha values so that the values of the other samples are condensed. Consider, perhaps using a logscale for the yaxis of the Fisher Alpha panel, alternatively the authors could exclude from this plot sample M77/2-776 and with a big red arrow just tell the reader the values of this 'outlier'. (Bottom and Middle panel) I assume the bars are 'errorbars' – some seem to be not symmetrical around the datapoint (possible depending on the statistic used) but more importantly Site 830, 1004P1 the error bars are below the datapoint.**

*We will check this with the software (PAST) and improve the figure accordingly.*

**Figure 5 to 7: Is it possible to have a negative value for oxygen concentration?**

*As previously discussed, no it is not possible to have negative concentrations. This is potentially the artefacts of the transfer function where the estimated values are biased towards lower values. We think that it is because majority of the samples are from really low oxygen concentration depths. Nevertheless, our estimations are positive within the statistical uncertainty. This is the reason we keep our discussion with changes over time but not exact values for specific time. We will add a sentence about this in section 4.2 where relevant.*

**Figure 5: give a 1:1 line**

*1:1 line for both graphs will be shown in the revised version*

**References**

Cardich, J., Gutiérrez, D., Romero, D., Pérez, A., Quipúzcoa, L., Marquina, R., Yupanqui, W., Solís, J., Carhuapoma, W., Sifeddine, A., and Rathburn, A.: Calcareous benthic foraminifera from the upper central Peruvian margin: control of the assemblage by pore water redox and sedimentary organic matter, Marine Ecology Progress Series, 535, 63-87, 10.3354/meps11409, 2015.

Clark JR, L. C., Wolf, R., Granger, D., and Taylor, Z.: Continuous recording of blood oxygen tensions by polarography, Journal of applied physiology, 6, 189-193, 1953.

Dale, A. W., Sommer, S., Lomnitz, U., Montes, I., Treude, T., Liebetrau, V., Gier, J., Hensen, C., Dengler, M., Stolpovsky, K., Bryant, L. D., and Wallmann, K.: Organic carbon production, mineralisation and preservation on the Peruvian margin, Biogeosciences, 12, 1537-1559, 10.5194/bg-12-1537-2015, 2015.

Erdem, Z., and Schönfeld, J.: Pleistocene to Holocene benthic foraminiferal assemblages from the Peruvian continental margin, Palaeontologia Electronica, 20.2.35A, 1-32, 2017.

Kalvelage, T., Lavik, G., Lam, P., Contreras, S., Arteaga, L., Löscher, C. R., Oschlies, A., Paulmier, A., Stramma, L., and Kuypers, M. M.: Nitrogen cycling driven by organic matter export in the South Pacific oxygen minimum zone, Nature geoscience, 6, 228, 2013.

Mallon, J., Glock, N., and Schönfeld, J.: The response of benthic foraminifera to low-oxygen conditions of the Peruvian oxygen minimum zone, in: Anoxia, Springer, Dordrecht, 305-321, 2012.

Martin, J. H., Knauer, G. A., Karl, D. M., and Broenkow, W. W.: VERTEX: carbon cycling in the northeast Pacific, Deep Sea Research Part A. Oceanographic Research Papers, 34, 267-285, 1987.

Pennington, J. T., Mahoney, K. L., Kuwahara, V. S., Kolber, D. D., Calienes, R., and Chavez, F. P.: Primary production in the eastern tropical Pacific: A review, Progress in Oceanography, 69, 285-317, 2006.

Revsbech, N. P., Larsen, L. H., Gundersen, J., Dalsgaard, T., Ulloa, O., and Thamdrup, B.: Determination of ultra-low oxygen concentrations in oxygen minimum zones by the STOX sensor, Limnology and Oceanography-Methods, 7, 371-381, 2009.

Schönfeld, J., Kuhnt, W., Erdem, Z., Flögel, S., Glock, N., Aquit, M., Frank, M., and Holbourn, A.: Records of past mid-depth ventilation: Cretaceous ocean anoxic event 2 vs. Recent oxygen minimum zones, Biogeosciences, 12, 1169-1189, 2015.

---

## Author Response (AR2)

**Response letter_2ⁿᵈ revision**

We appreciate the remarks and suggestions of the reviewers and are grateful for the effort the they have invested. We have modified the text substantially in response to reviewer comments. Below we respond to each comment individually and indicate the revisions accordingly. For clarity, referee comments are indicated in bold and Authors' comments are indicated in italics. All the changes in the revised manuscript are highlighted with yellow.

Reply to Anonymous Referee #2:

**METHODS:**

**In the manuscript, the authors still do not present arguments why a relationship of living benthic foraminifera abundance with dissolved oxygen concentration can be applied to a dead assemblage. In the response to reviewers comments the authors agree there are substantial differences between the two populations.**

*It seems like there is a misunderstanding. We cannot compare living (rose Bengal stained) benthic foraminiferal faunas and dead (empty tests, not stained) benthic foraminiferal assemblages from surface sediment samples because the respective census data are not available from any of the Recent distributional studies in the Peruvian OMZ which are considered in this paper. In the revised version, we indicated this under section 2.2 by adding a text. Living assemblages are the best means to correlate with prevailing environmental conditions measured at the time of collection, whereas the dead assemblages are an integral of many generations and may include specimens that were living under different conditions than those measured at the time of collection. In most of the surface samples, however, and in particular in those from the upper OMZ, nearly all benthic foraminiferal specimens were stained. With the given high accumulation rates, it appears to be less likely, that any dead assemblage will contain old, relictic specimens. Therefore, it is justified to assume that 1) living and dead foraminifera assemblages are not that different from each other in our samples, that 2) the dead assemblage accurately mirrors the prevailing environmental conditions at which the living fauna was thriving, and 3) that only well-constrained taphonomical bias inferred an alteration of the dead assemblage during fossilisation (see below).*

*To eliminate the confusion in the rest of this response letter, we will use the following words for different groups of foraminifera mentioned in this study: "living" for stained specimens and "dead" for specimens that were not stained in surface sediments, whereas "fossil" will refer to benthic foraminifera observed in the long sediment cores (Table 3 of the MS; M77/2-50-4, 52-2 and 59-1).*

*In our first response letter, we indeed mentioned a comparison between living species and fossil ones observed in the cores. It has to be emphasized, however, that a reduced living benthic foraminiferal dataset (the most abundant 16 species) is not identical with the abundant species (>5%) observed in at least one of the three sediment cores (see Table 1 below). In particular there is one species, Cancris carmenensis, is absent from any samples of concerned time intervals in any sediment core. The remaining 15 species are indeed observed in sediment cores. Seven of these species (indicated by bold in Table 1 below) are common with >5% in relative abundance in at least one sediment core. We hypothesize this substantial difference between living reference dataset and the fossil dataset reflects differences in paleoenvironmental conditions. Considering where sediment cores were obtained and how surface sediment samples distributed (Figure 2 of the MS), such a contrast in species lists is not*

surprising. It is also not to wonder that this discrepancy is potentially reflected in the transfer function, estimated values and errors. It is possible that the overall estimated values are somewhat low, because the majority of reference surface samples are from depths shallower than the sediment cores investigated, hence at lower oxygen concentrations. However, without additional samples and further study, we see no way to correct this potential bias. If we knew that a particular species appears only above 45 μmol/kg we could potentially correct estimated by taking that value as a baseline or check point. However, the limited understanding of benthic foraminiferal species relationships with bottom water characteristics does not enable us to make such assumptions. In our opinion, to make such specific assertions about species oxygen tolerances would be wrong and misleading. Therefore, we abstain from using the estimates at face value but rather take the more conservative and realistic approach of using changes through time as calculated for each sediment core and time period. We are aware of the limitations and indicated so in the text. This approach is very new, and we believe that it is a worthwhile methodology to pursue. As with any new approach, precision will improve as additional samples from mid-depths of the Peruvian Margin and more species' information becomes available.

Table 1. The most abundant species (>5%) observed in both surface sediments (reduced living benthic foraminifera dataset) and in downcore records (fossil assemblages observed sediment cores considering the last 25 kyr).

| Living dataset 16 species | in cores | | "Fossil" >5% in at least one core |
|---|---|---|---|
| **Bolivina costata** | common | | Alabaminella weddellensis |
| Bolivina interjuncta | | | Anomalinoides minimus |
| Bolivina plicata | | | **Bolivina costata** |
| Bolivina seminuda | | | Bolivina interjuncta var. bicostata |
| **Bolivina spissa** | common | | Bolivina pacifica |
| **Bolivinita minuta** | common | | Bolivina quadrata |
| Cancris carmenensis | absent | | Bolivina seminuda var. humilis |
| Cassidulina crassa | | | **Bolivina spissa** |
| **Cassidulina delicata** | common | | **Bolivinita minuta** |
| **Epistominella obesa** | common | | Buccella peruviana |
| **Epistominella pacifica** | common | | Bulimina exilis |
| Fursenkoina fusiformis | | | Bulimina pagoda |
| Gyroidina soldanii | | | Cassidulina auka |
| Suggrunda porosa | | | Cassidulina carinata |
| **Uvigerina peregrina** | common | | **Cassidulina delicata** |
| Valvulineria glabra | | | Cassidulina laevigata |
| | | | Cassidulina minuta |
| | | | Cibicides mckannai |
| | | | Epistominella afueraensis |
| | | | Epistominella exigua |
| | | | **Epistominella obesa** |
| | | | **Epistominella pacifica** |
| | | | Epistominella smithi |
| | | | Fursenkoina cornuta |
| | | | Gyroidina rothwelli |
| | | | Pseudoparella subperuviana |
| | | | Pseudoparella sp. |
| | | | Uvigerina auberiana |
| | | | Uvigerina bifurcata |
| | | | **Uvigerina peregrina** |
| | | | Uvigerina semiornata |
| | | | Virgulina spinosa |

**ACTION:**

**The authors need to include statements in their manuscript highlighting that the dead and living populations are intrinsically different at the sites, and discuss how this influences the results. For example, could this explain the huge differences in dissolved oxygen in Holocene versus present day oxygen concentrations? Or do the authors think there has been a huge improvement over the last ca 5 kyr (which would be a big thing?) Either way this needs to be thoroughly explained in the main text.**

*We hope that we have cleared up the misunderstanding on dead vs living foraminifera contrasts. Reasons for relatively low Holocene values vs present values will remain elusive until additional work is undertaken. As indicated in previous comments, the low estimated values of this time frame may or may not be an artefact of the limited data set. Thus, bottom waters might be more oxygenated at the later stage of the early Holocene. Since there are not many quantitative investigations on that time scale allowing a comparison, and the scope of our observations is limited, we cannot further comment on this. Additional samples and study are needed to confirm or modify the paleoceanographic implications of our results. This issue is now detailed in the Discussion of the revised manuscript.*

**Would it not make more sense to develop a relationship between recent dead populations with dissolved oxygen concentrations to apply down-core? If the authors believe this is not a valid approach, this should be discussed in the manuscript.**

*We cannot proceed with such an approach because this information is not available. We argue that using dead populations rather than living populations introduces significant artefacts into a transfer function. As noted by a number of workers, wherever possible, it is best to use specimens that were living in the environmental conditions measured at the time of collection. We added text about this under section 2.2. Nevertheless, since the majority of the specimens in surface sediments were stained, it is not likely that there would be a substantial difference in the statistical analysis if the dead assemblage was used.*

**I am not sure what the difference is between making quantitative statements about specific periods compared with making quantitative statements at one location between specific periods?**

*We perceive this comment as the reviewer rather quotes for describing the temporal variability at a standard record and to abstain from a differentiated consideration of each locality. The difference of the proxy values among the sediment cores accounts for a spatial variability in oxygenation, as it is observed today and most likely also prevailed in the geological past. Today's spatial variability in oxygenation on the scale of several hundreds of kilometres and hundreds of metres in depth is affected by an interplay between surface ocean productivity and the remineralisation of particulate organic matter at depth, and the advection of more oxygenated waters from the North. Therefore, any variability in hydrographic or chemical properties through time, e.g. oxygen or temperature, has to be regarded in a spatial context. It is a given challenge of paleoceanographic studies in general to approach an understanding or at least the recognition of this dynamics. We have discussed this already in an earlier paper (Erdem et al., 2016), to which reference is given in the revised version of the manuscript.*

**How were standard deviations calculated?**

*Below we describe the calculation in detail. This information will be provided in supplementary material.*

*The polynomial transfer function is given in the form of eq.1:*

Eq.1: $$x = C + \sum RCo_n \cdot \%_n$$

*where x is the environmental variable that should be reconstructed with the transfer function (in this case $[O_2]_{BW}$ or RRPOC); C is a constant from the multiple regression (tab. 5); $RCo_n$ is the regression coefficient for the foraminiferal species n (tab. 5); and $\%_n$ is the percentage of the foraminiferal species n within the assemblage.*

*For the calculation of the errors for x (i.e. $[O_2]_{BW}$ or RRPOC) a complete error propagation has been done including the 1σ errors of all species within the polynomial transfer function. The error propagation has been applied to the polynomial transfer function (Eq. 1) in the form of equation Eq. 2:*

Eq.2: $$\sigma_x = \sqrt{\left(\frac{\partial x}{\partial C} \cdot \sigma_C\right)^2 + \sum\left(\frac{\partial x}{\partial RCo_n} \cdot \sigma_{RCo_n}\right)^2 + \sum\left(\frac{\partial x}{\partial \%_n} \cdot \sigma_{\%_n}\right)^2}$$

*where $\sigma_x$ is the standard deviation (1sd) of the environmental variable x (i.e. $[O_2]_{BW}$ or RRPOC) $\sigma_C$ is the error of the constant C (tab.5); $\sigma_{RCo_n}$ is the standard error for the regression coefficient RCo for the foraminiferal species n (tab.5); and $\sigma_{\%_n}$ is the standard error for $\%_n$. Solution of the derivatives in Eq.2 results in Eq.3:*

Eq.3: $$\sigma_x = \sqrt{\sigma_C{}^2 + \sum\left(\%_n \cdot \sigma_{RCo_n}\right)^2 + \sum\left(RCo_n \cdot \sigma_{\%_n}\right)^2}$$

*We have to state that we neglect the last term in equations 2 and 3 because we do not know $\sigma_{\%_n}$, since three replicates would be necessary for each sample to determine this error. In this study all downcore samples were counted by a single investigator. Previous studies showed that population densities of sample replicates, which have been picked dry by a single investigator had an accuracy (1σ) of ±2 % (Schönfeld et al., 2013). Thus, in comparison with the high $\sigma_C$ and $\sigma_{RCo_n}$ (tab.5) the error $\%_n$ should be negligible. Nevertheless, the proportions of frequent species within the same assemblage may differ by 2–7 % between different picking modes or laboratories (Schönfeld et al., 2013).*

**AGE MODEL:**

**The information provided is not satisfactory.**

**ACTION:**

**The authors need to visually show how benthic d18O records were correlated to stacked records of Antarctic ice cores (this information is not in the referenced publications). I am especially curious to find out how this was done for the Holocene and how the early and mid Holocene were differentiated for core 52-2. In addition, the authors need to describe how radiocarbon ages and oxygen isotope dates were determined within the hiatuses of 47-2 and 50-4.**

**For the shallow core, there are only two 'ages' from the late deglaciation and early Holocene. How can you confidently constrain ages between 13 and 22, including the LGM interval?**

*Following both reviewers' suggestion and in order to improve this issue, we decided to skip the record of the shallow core 47-2 for the time being. Accordingly, all the information on core 47-2 is removed from figures and text of the revised version.*

**The authors need to show the evidence (d18O graph for each site and correlation to which record) in the supplementary information. Stern and Lisiecki (2014) provide stacked records for different ocean basins and water depths, it is more appropriate to use these for correlation purposes.**

*We are following the suggestion and we add another figure introducing the age model of concerned sediment cores. The figure is originally published as supplementary material to Erdem et al. (2016), for this publication we modified it and added additional information, in particular benthic stack record from Stern and Lisiecki (2014). We used Pacific Intermediate water benthic isotope stack (supplementary dataset stored in Pangaea (Stern and Lisiecki, 2018)) for correlation with the sediment cores concerned. Indeed, we realized that a better fine tuning was necessary, particularly for the deglaciation section of the core 52-2 (see below graph, Fig. 1). We did not tune the other two core records (50-4 and 59-1) as their age models were seemingly better established. Figure 6 and 7 of the manuscript are modified accordingly considering the new age model.*

*The reservoir ages at our sampling locations likely varied over the last deglaciation. Unfortunately, data about changes in reservoir ages is scarce. The closest records of changing planktic $^{14}$C reservoir ages are located in the equatorial Pacific, close to Panama (Zhao and Keigwin, 2018) and off Chile (Sarnthein et al., 2019; Siani et al., 2013). These $^{14}$C reservoir ages differ significantly from each other during some periods and likely cannot be applied to our study sites. Further uncertainty is added to cal. ages based on planktic $^{14}$C ages by distinct plateaus in atmospheric $^{14}$C ages that can last for up to 1,000 yr (Sarnthein et al., 2015, 2019). Therefore, it is not possible to assign rapid short time changes to distinct short events in our records and we focus more on the long-term trends in our record. The reservoir ages are likely not significantly different between our three study sites, due to their regional proximity. Thus, the relative trends between our sediment records are supposed to be valid. Future studies on changing $^{14}$C reservoir ages in the ocean will improve age models of existing paleorecords.*

*The current age model of core 52-2 is based on five 14C dating, three tie points by correlation to core 59-1 for the Holocene part, three new tie points by correlation to benthic isotope stack for the deglaciation period and three tie points to EPICA ice core for the later part of the record. Overall, sedimentation rates throughout the core is relatively stable (around 30 cm/kyr).*

[Figure]

*Figure 1. Before (orange) and after (purple) fine tuning of core 52-2 age model by correlating benthic $\delta^{18}O$ from Stern and Lisiecki (2014) with $\delta^{18}O$ of core 52-2.*

**Other comments and observations that need actioning:**

**In the abstract it is stated that 'Each core displayed a similar trend of decreasing oxygen levels since the LGM.' This is not true, the shallow core is an exception.**

*Shallow core (M77/2-47-2) is removed from this manuscript due to reasonable concerns on the validity of the age model (see above). The Abstract is modified accordingly.*

**The statement about the changes being time transgressive is strange and does not seem corroborated by the data. The shallower cores (at 1013 m and 997 m from 8 degrees South and 4 degrees South) start changing around the same time, whereas the deeper core seems to change later. In addition, for the deeper core the age model is based on two AMS ages around 13 and 22 cal ka, so exact timing of change is not constrained.**

*With the new tuned age model for the deep core (52-2) this pattern is a bit more visible. However, we now do not have information on the full HS1 period. Considering the comments from both reviewers we modified our interpretation and limit the speculative tone of discussion on time transgressive expansion of the OMZ. Accordingly, we removed a substantial part from section 4.2. Interpretation of the results also changed. All new sentences are highlighted with yellow.*

**Does the last sentence of the abstract refer to data in the article? If not it needs a reference.**

*Considering the modifications in the interpretation, discussion part, we removed the last sentence from the abstract.*

**Page 13 1st sentence: what does this mean, 'a slight recovery of the OMZ'? Did it become less or more oxygenated?**

*More oxygenated. We rephrased this section accordingly.*

**Figure 1 and 2 need legends for oxygen concentrations.**

*Done*

**What does Figure 4 show? Discuss in detail in the main text.**

*With the age model figure included we decided to move this figure to supplementary information. A broader explanation is given in supplementary by comparison of CCA application to census dataset and reduced dataset.*

**What is the purpose of Table 1? Generally oxygen threshold are related to redox reactions, not biota abundance.**

*The threshold values mentioned in this table have been used in benthic foraminifera related publications. We decided to include this comparison in order make aware of current classifications used in the literature and their differences.*

Reply to Anonymous Referee #3:

**The authors have detailed how to account for different assemblages downcore than the living ones. But I think the reviewers were also concerned on changes in preservation through time. The depositional setting off Peru is pretty harsh towards the preservation of foraminiferal shells and changes a lot through time, i.e. with the changes in upwelling. So the assemblages may be biased by dissolution of more dissolution-susceptible species.**

**Additionally to this, I have two further points, firstly Bengal Rose was used to identify living specimens, but especially in low-oxygen settings this may largely over-estimate the population of living specimens;**

*We used a conservative approach to assess rose Bengal stained specimens, which has been shown to be a reasonable means to evaluate living populations (Murray and Bowser, 2000; Schönfeld, 2012). This staining method is also widely used in benthic foraminifera studies from oxygen minimum settings to determine the living specimens (e.g., (Jannink et al., 1998; Caulle et al., 2014; Koho et al., 2015). Since the work of Figueira et al. (2012), it is now generally accepted.*

**and secondly, I would like to see some information on how oxygen changes at these core locations through the year, i.e. seasonal changes. Could this be a reason for the scatter or apparent offset between reconstructed values and values at the time of sampling? I.e. the forams may have been alive during sampling but maybe the conditions were not ideal such that they were "hibernating" and actually have their active life phase during another time of the year with different oxygen concentrations.**

*Bottom water oxygen concentrations around the lower boundary in the region is relatively stable throughout the year whereas the upper boundary is quite dynamic (Paulmier and Ruiz-Pino, 2009). Our living benthic foraminifera compilation dataset was reduced to stations below 300m water depth. Multiple regression used for the downcore application concerns only deeper sampling locations therefore living foraminifera from these deeper than 300m water depth are not expected to experience seasonal $[O_2]_{BW}$ variability as much as stations along the shelf (50-100m) would experience. Nevertheless, we compared the differences between our datasets and between each other for deeper stations since sampling period and seasons showed differences (Perez et al., sampled during December – January 1998; M77 took place during October to December in 2008 and latest expedition M137 was in May 2013). We compared the $[O_2]_{BW}$ from stations close to each other (e.g., around 12°S similar depths) and the only difference was observed at two 820 m stations, the other stations did not show significant contrast between different sampling periods (Table 4 in the MS), the other stations did not show significant contrast between different sampling periods. Therefore, we consider the living assemblage data as being representative and in equilibrium with the prevailing environmental conditions. We added text about this under section 2.2.2.*

**The age model is indeed a necessary point to be added. I agree that it appears that there is a prograding trend in changing oxygen values but unless the age models are very precise I would tone this down a bit.**

*We are following the suggestion and we add another figure introducing the age model of concerned sediment cores as mentioned earlier (see above).*

**The data used in this manuscript should be reported as a dataset in for example Pangaea. Currently there are only references to the different papers where assemblage, core top and water data have been presented.**

*The datasets are currently stored in Pangaea. Upon publication they will be publicly available.*

**References:**

Caulle, C., Koho, K. A., Mojtahid, M., Reichart, G. J., and Jorissen, F. J.: Live (Rose Bengal stained) foraminiferal faunas from the northern Arabian Sea: faunal succession within and below the OMZ, Biogeosciences, 11, 1155-1175, 10.5194/bg-11-1155-2014, 2014.

Erdem, Z., Schönfeld, J., Glock, N., Dengler, M., Mosch, T., Sommer, S., Elger, J., and Eisenhauer, A.: Peruvian sediments as recorders of an evolving hiatus for the last 22 thousand years, Quaternary Science Reviews, 137, 1-14, 10.1016/j.quascirev.2016.01.029, 2016.

Figueira, B.O., Grenfell, H.R., Hayward, B.W. and Alfaro, A.C.: Comparison of Rose Bengal and CellTracker Green staining for identification of live salt-marsh foraminifera. *The Journal of Foraminiferal Research*, *42*(3), 206-215, doi.org/10.2113/gsjfr.42.3.206, 2012.

Jannink, N. T., Zachariasse, W. J., and van der Zwaan, G. J.: Living (Rose Bengal stained) benthic foraminifera from the Pakistan continental margin (northern Arabian Sea), deep-sea research I, 45, 1483—1513, 1998.

Koho, K., de Nooijer, L., and Reichart, G.: Combining benthic foraminiferal ecology and shell Mn/Ca to deconvolve past bottom water oxygenation and paleoproductivity, Geochimica et Cosmochimica Acta, 165, 294-306, 2015.

Murray, J. W., and Bowser, S. S.: Mortality, protoplasm decay rate, and reliability of staining techniques to recognize 'living' foraminifera: a review, Journal of Foraminiferal Research, 30, 66-70, 2000.

Paulmier, A., and Ruiz-Pino, D.: Oxygen minimum zones (OMZs) in the modern ocean, Progress in Oceanography, 80, 113-128, 2009.

Sarnthein, M., Balmer, S., Grootes, P. M. and Mudelsee, M.: Planktic and Benthic 14C Reservoir Ages for Three Ocean Basins, Calibrated by a Suite of 14C Plateaus in the Glacial-to-Deglacial Suigetsu Atmospheric 14C Record, Radiocarbon, 57(1), 129–151, doi:DOI: 10.2458/azu_rc.57.17916, 2015.

Sarnthein, M., Küssner, K., Grootes, P. M., Ausin, B., Eglinton, T., Muglia, J., Muscheler, R. and Schlolaut, G.: Plateaus and jumps in the atmospheric radiocarbon record – Potential origin and value as global age markers for glacial-to-deglacial paleoceanography, a synthesis, Clim. Past Discuss., 2019, 1–63, doi:10.5194/cp-2019-127, 2019.

Siani, G., Michel, E., De Pol-Holz, R., Devries, T., Lamy, F., Carel, M., Isguder, G., Dewilde, F. and Lourantou, A.: Carbon isotope records reveal precise timing of enhanced Southern Ocean upwelling during the last deglaciation, Nat. Commun., 4(May), 1–9, doi:10.1038/ncomms3758, 2013.

Schönfeld, J.: History and development of methods in Recent benthic foraminiferal studies, Journal of Micropalaeontology, 31, 53-72, 2012.

Stern, J. V., and Lisiecki, L. E.: Termination 1 timing in radiocarbon-dated regional benthic δ18O stacks, Paleoceanography, 29, 1127-1142, 2014.

Stern, J.V; Lisiecki, L., E: Regional benthic $\delta^{18}$O stacks and their d18O uncertainties. PANGAEA, https://doi.org/10.1594/PANGAEA.891137, 2018.

Zhao, N. and Keigwin, L. D.: An atmospheric chronology for the glacial-deglacial Eastern Equatorial Pacific, Nat. Commun., 9(1), 3077, doi:10.1038/s41467-018-05574-x, 2018.

---

## Author Response (AR3)

Reply to Anonymous Referee #2 for final comments:

We would like to thank referee 2 for his/her careful reviews. We are grateful for the effort he/she has invested. Whole manuscript was read once more and minor mistakes are now corrected including the ones reviewer 2 pointed out.

Regarding the dataset: the compilation dataset (rose Bengal stained living benthic foraminifera) is already on Pangaea database and can be reached through the following link:

[https://doi.pangaea.de/10.1594/PANGAEA.901840](https://doi.pangaea.de/10.1594/PANGAEA.901840)

Upon publication the dataset will be open to public. It is also linked to radiocarbon ages of the sediment cores therefore all the information concerned for this study will be easily accessible.